# ESPACE: Dimensionality Reduction of Activations for Model Compression

**Charbel Sakr**
NVIDIA Research
csakr@nvidia.com

**Brucek Khailany**
NVIDIA Research
bkhailany@nvidia.com

## Abstract

We propose ESPACE[1], an LLM compression technique based on dimensionality reduction of activations. Unlike prior works on weight-centric tensor decomposition, ESPACE projects activations onto a pre-calibrated set of principal components. The activation-centrality of the approach enables retraining LLMs with no loss of expressivity; while at inference, weight decomposition is obtained as a byproduct of matrix multiplication associativity. Theoretical results on the construction of projection matrices with optimal computational accuracy are provided. Experimentally, we find ESPACE enables 50% compression of GPT3, Llama2, and Nemotron4 models with small accuracy degradation, as low as a 0.18 perplexity increase on GPT3-22B. At lower compression rates of 20% to 40%, ESPACE drives GPT3 models to outperforming their baseline, by up to a 0.38 decrease in perplexity for GPT3-8B. ESPACE also reduces GEMM execution time and prefill inference latency on existing hardware. Comparison with related works on compressing Llama2-7B via matrix factorization shows that ESPACE is a first step in advancing the state-of-the-art in tensor decomposition compression of LLMs.

## 1 Introduction

Capabilities of large language models (LLMs) have recently soared in natural language understanding and generative power. It is appreciated that there exists a correlation between model size and achievable accuracy. Indeed, as LLMs consume trillions of tokens during their training, a large parameter volume is required to capture intricate linguistic features [1]. This leads to a trade-off in LLMs: larger parameter counts improve accuracy but come with increased serving cost.

However, it is also appreciated that the computational requirements of inference may be lower than those of training [2]. To that end, numerous studies have investigated compression of LLMs to reduce inference cost. The most popular LLM compression techniques are quantization [3] and pruning [4]. A less explored, but powerful technique is *tensor decomposition*, and in our work, we propose a novel, *activation-centric* way to decompose LLM tensors. Our proposal is to project activations onto a static set of components optimizing fidelity. The projection reduces activation dimensionality and leads to weight compression at inference as a byproduct of matrix multiplication associativity.

### 1.1 Related work and motivation for activation-centric tensor decomposition

Recent research has proposed many **quantization** and **pruning** techniques for compressing LLMs. Examples of advances in LLM quantization include SmoothQuant [3], AWQ [5], and GPTQ [6]; while notable LLM pruning works include SparseGPT [4], LLM-Pruner [7], and ReLU-based masking [8]. These methods are conceptually orthogonal to our proposal for activation projection which can be

---

[1]We use the french pronunciation "espace", which means "space".

38th Conference on Neural Information Processing Systems (NeurIPS 2024).

implemented in low precision or sparse formats. Nevertheless, compression fundamentally introduces noise, and an open problem is to study the impact of combining different methods, e.g., quantization and matrix factorization. This is beyond the scope of our paper, but a good direction for future work.

Our work is also orthogonal to **non-compressive** LLM serving acceleration such as continuous batching [9] or speculative decoding [10], and attention optimizations such as PagedAttention [11], RadixAttention [12], and FlashAttention [13]. Our study is on matrix multiplication layers involving weights and activations, and hence is mutually exclusive to works improving cross multiplications of activation tensors in attention. In fact, all our experiments use FlashAttention.

Finally, we turn to **tensor decomposition**, also known as **factorization**. Thus far, compression for LLM inference using factorization has been focused on **weight decomposition**. KnGPT [14] uses the Kronecker transform to pack a large matrix into two smaller ones. TSVD [15] performs iterative singular value decomposition (SVD) on weight matrices to produce high rank ternary components. TensorGPT [16] and HEAT [17] compress weight matrices into a cascade product of small matrices using the tensor-train algorithm. SVD-LoRa [18] uses a truncated SVD on weights and finetunes the model using LoRa [19]. The LoRa adapters are then merged to the main branch using bounds on the rank of sum of low rank matrices. ASVD [20] performs a truncated SVD on the weights after re-scaling them by a diagonal matrix and their inverse encapsulating activation statistics. This work realizes the importance of activation-awareness but still uses weight-centric compression. SliceGPT [21] extracts principal components in normalization layers to guide the deletion of rows and columns in weight matrices. The compression is achieved using a factorization made implicit via computational invariance. The statistical method employed by sliceGPT shares similarities with one of our results, but our problem formulation and solution are different.

Factorization can also streamline LLM training and finetuning. For instance, LoRa [19] finetunes pre-trained models using residual low rank adapters which are then absorbed into the main branch. Similarly, GaLore [22] applies a low rank approximation to gradients in back-propagation. These works do not modify inference parameter and operation count, and are hence orthogonal to ours. Our method could be applied in tandem with LoRa or GaLore, but this is beyond the scope of this paper.

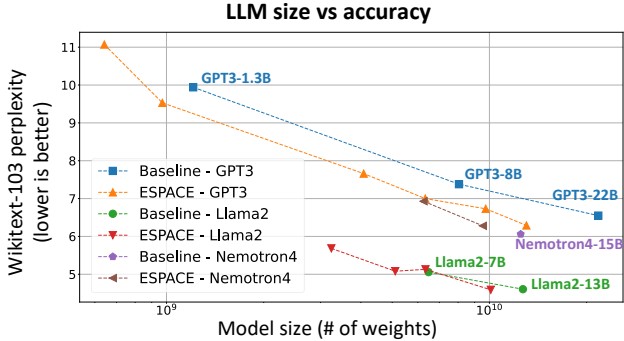

Figure 1: Perplexity[2] versus model size for GPT3 and Llama2 models and comparison to compressed models using ESPACE.

Since factorization increases the number of LLM tensors, achieving high compression rates requires the intermediate dimensions to be much smaller than that of original dot product. This breakage in computation usually necessitates a retraining or finetuning stage to be healed. Unfortunately, this healing process is impeded because factorized LLMs have fewer learnable parameters which decreases expressivity [17, 22].

To our knowledge, no prior art has explored **activation decomposition**. Indeed, applying factorization solvers (e.g., SVD) dynamically incurs large inference runtime overheads. Yet, activation decomposition has several desired features which we examine in Section 2 and motivate via the following insights: (a) weights stay uncompressed during retraining, preventing the aforementioned loss of expressivity; (b) large activation tensors contain inherent redundancies making them prime candidates for compression; and (c) since most LLM computation comprises multiplications of weights and activations, decomposing the latter can lead to compressing the former at inference.

## 1.2 Contributions

We propose Eigen Static Principal Activation Component Estimation (ESPACE), an LLM compression technique based on activation dimensionality reduction. Our contributions are as follows:

- We project activation tensors onto a static and pre-calibrated orthonormal matrix. The projection lowers activation dimensionality but keeps weight matrices intact and fully available for training.

At inference, leveraging matrix multiplication associativity, model compression is achieved through pre-computation of the product of weight and projection matrices.

- We theoretically derive optimal constructions for activation dimensionality reduction. Specifically, the projection matrix is calibrated in a manner to minimize activation decomposition mean squared error and forward propagated noise metrics. The solution is based on an eigenvalue decomposition of activation auto-correlation and yields multiple candidate projections for each activaton tensor.

- We empirically study compression of models in the GPT3, Llama2, and Nemotron4 families evaluated on the Wikitext-103 dataset for perplexity and the LM evaluation harness for downstream task accuracy. The amelioration in size versus perplexity[2] trade-offs is summarized in Figure 1.

- We show that ESPACE can compress LLMs by ∼50% at the cost of a small accuracy loss, as low as 0.18 increase in perplexity on GPT3-22B.

- At lower compression rates, we find encouraging empirical evidence that ESPACE filters out noise and improves accuracy; e.g., ∼20% compressed GPT3-8B lowers its baseline perplexity by 0.38.

- As an additional benefit of ESPACE, tangible latency reduction of 35%-to-45% is obtained in matrix multiplication layers. This speed-up translates to up to ∼ 40% faster prefill inference latency metricized by the time to first token and measured on existing hardware.

- By comparison to existing works on tensor decomposition, we determine that ESPACE is a first step in pushing the frontier of compression rate versus accuracy retention (see Figure 4).

## 2 Dimensionality Reduction & Projections

In this section, we introduce notation for matrix multiplication, review weight decomposition, and introduce our proposed mechanism of dimensionality reduction via activation projections.

### 2.1 Matrix Multiplication and Weight Decomposition

We consider general matrix multiplications (GEMMs) described in Figure 2(a) of the form

$$\mathbf{Y} = \mathbf{W}^T\mathbf{X} \tag{1}$$

where $\mathbf{W}$ is a weight matrix of size $K \times N$ and $\mathbf{X}$ is an input activation tensor of size $K \times M$ so that the output activation tensor $\mathbf{Y}$ is of size $N \times M$. Typically, $K$ and $N$ are defined by network topology and layer instance, they are commonly referred to as *embedding* or *hidden* size. In contrast, $M$ stacks multiple dimensions in an activation tensors to obtain a 2D matrix view. Generally, these are the *sequence* and *batch* dimensions.

Transformer-based LLMs have four GEMM layers per block: query-key-value (QKV), projection (Proj), fully connected 1 (FC1), and fully connected 2 (FC2) layers. Our study is concerned with these layers, while cross activation multiplication and embedding layers are untouched. For notational simplicity, in this paper, *we do not include layer indices in our equations*.

The matrix $\mathbf{W}$ in (1) stores layer parameters and dictates the model's inference accuracy. To improve convergence of these parameters, an optimizer state is stored alongside weights during training and tracks historical values of gradients and updates [23, 24]. On the other hand, the activation tensor $\mathbf{X}$ depends on the input stimulus to the network, and is therefore generated on the fly.

Thus, at inference, weights are fixed but activations are dynamic. As a consequence, prior work on tensor decomposition has focused on compressing frozen weight matrices. One way of doing so is breaking $\mathbf{W}^T$ into a low-rank approximation using some form of truncated SVD [20, 18], which is described in Figure 2(b). Specifically, (1) is approximated as:

$$\mathbf{Y} \approx \mathbf{UVX} \tag{2}$$

where $\mathbf{U}$ and $\mathbf{V}$ are matrices of size $N \times L$ and $L \times K$, respectively, with $L$ being the factorization rank. For the decomposition procedure to be useful, two conditions need to be met: (a) $L << \min(K, N)$ for compression, and (b) the approximation $\mathbf{W}^T \approx \mathbf{UV}$ should be accurate. However, achieving both conditions simultaneously may be challenging because a very low rank factorization usually leads to significant accuracy drop [22].

---

[2]Perplexity depends on tokenizer so that comparisons across LLM families (GPT3 vs Llama2) are not useful.

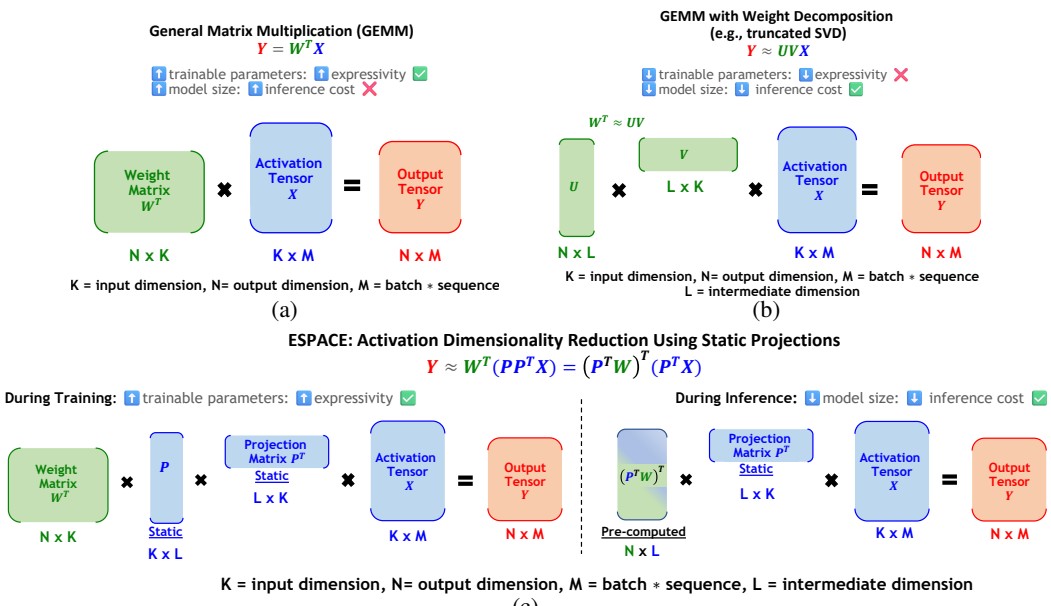

Figure 2: Decompositions in GEMMs: (a) baseline multiplication of weight matrix and activation tensor, (b) truncated SVD on the weight matrix, and (c) proposed approach of inserting a static matrix to project activations. With ESPACE, all weights are available for training, while inference compression is achieved via per-computation of $(\mathbf{P}^T\mathbf{W})$.

As with other compression techniques, e.g., quantization and pruning, retraining of the compressed model may be employed to recover accuracy. However, the decomposition in (2) introduces two training-related hurdles: (a) the effective number of trainable parameters has decreased significantly which reduces model expressivity, and (b) the breakage of spatial weight structure prevents the retraining procedure from loading the original optimizer state. Retraining a model without its optimizer state is known to introduce significant difficulty in convergence [17].

## 2.2 Activation Decomposition via Static Projection

Since weight decomposition poses the above hurdles, we motivate the need for an activation-centric solution. In some measure, activation compression may be more achievable due to the large stack dimension $M$ comprising batches and sequences. Statistically, the Central Limit Theorem claims that stacking data is likely to exhibit redundancies [25]. In the case of LLMs, such redundancies are further pronounced due to the likelihood of repeated tokens and information in natural language.

Therefore, activations should be prime candidates for tensor decomposition. Nevertheless, prior arts have not explored activation decomposition due to one fundamental limitation: unlike weights, activations are generated on the fly; meaning that tensors must be compressed during inference, potentially incurring large runtime penalties.

We propose to apply *static* dimensionality reduction on the activation tensor $\mathbf{X}$ in (1). Concretely, our proposal is to project $\mathbf{X}$ onto a pre-computed static orthonormal matrix $\mathbf{P}$ of size $K \times L$, where crucially $L << K$. Reconstructing $\mathbf{X}$ requires a re-expansion using the transpose of the projection matrix, i.e., $\mathbf{X} \approx \mathbf{P}\mathbf{P}^T\mathbf{X}$. While $\mathbf{P}^T\mathbf{P} = \mathbf{I}_{L \times L}$, we note that $\mathbf{P}\mathbf{P}^T \neq \mathbf{I}_{K \times K}$ since $L << K$. Thus, the proposed activation transformation is noisy, and in Section 3, we derive optimal conditions on the calibration of $\mathbf{P}$ to minimize the effects of this noise.

Our proposal, described in Figure 2(c) is to approximate the GEMM in (1) using the following:

$$\mathbf{Y} = \mathbf{W}^T\mathbf{X} \approx \mathbf{W}^T\mathbf{P}\mathbf{P}^T\mathbf{X} = \mathbf{W}^T\left(\mathbf{P}\mathbf{P}^T\mathbf{X}\right) = \left(\mathbf{P}^T\mathbf{W}\right)^T\left(\mathbf{P}^T\mathbf{X}\right) \qquad (3)$$

where we used associativity of matrix multiplication to highlight key aspects of our approach.

**During training/finetuning:** we view our GEMM as $\mathbf{W}^T\left(\mathbf{P}\mathbf{P}^T\mathbf{X}\right)$ where $\mathbf{X}$ has been replaced by its approximation. We emphasize that $\mathbf{P}$ is static and does not get updated during training.

Meanwhile, $\mathbf{W}^T$ is fully available for adaptation to the activation approximation. The availability of all learnable weights elides losing model expressivity. The structure of $\mathbf{W}^T$ is also unchanged and can be mapped to the baseline's optimizer state. Thus, the proposed approach does not suffer from the same limitations as weight decomposition techniques. We do note that introducing $\mathbf{P}$ induces a small storage overhead at train time. However, when $L << K$, and the order of computation is properly compiled, the number of operations per iteration is lower than baseline training; and, though not central to our contribution, we did observe up to 15% reduction in training iteration time for 50% compressed models.

**During inference:** we view our GEMM as $\left(\mathbf{P}^T\mathbf{W}\right)^T\left(\mathbf{P}^T\mathbf{X}\right)$ where the required matrices are $\mathbf{P}$ of size $K \times L$ and $\left(\mathbf{P}^T\mathbf{W}\right)$ of size $N \times L$, which is pre-computed before deployment. Thus, per-layer parameter count required for inference has decreased from $KN$ to $L(K + N)$, which, provided $L << \{K, N\}$ presents an opportunity for significant model compression. For instance, if $N = K$, i.e., $\mathbf{W}^T$ is square, and $L = {}^K/4$, then our method yields 50% compression at inference time. This is one of the compression rates we target in Section 4.

We emphasize that $\mathbf{P}$ is not shared across GEMM layers; rather, each GEMM layer decomposed according to (3) has its own pre-calibrated matrix $\mathbf{P}$. Furthermore, by virtue of (3) not introducing dependencies across mini-batches, our method is fully compatible with data parallelism.

## 3 Eigen Static Principal Activation Component Estimation

Our proposed activation decomposition induces an approximation error as $\mathbf{X} \neq \mathbf{P}\mathbf{P}^T\mathbf{X}$. In this section, we first introduce an ergodic estimation of activation auto-correlation. This important statistic is then used for theoretical constructions of $\mathbf{P}$ with guarantees on computational accuracy. Multiple results are presented and then combined in our compression studies in Section 4.

### 3.1 Activation auto-correlation estimation

Let $\mathbf{x}$ be an arbitrary $K$-dimensional vector in $\mathbf{X}$; we define the *activation auto-correlation* matrix of size $K \times K$ as $\mathbf{C}_{\mathbf{X}} = \mathbb{E}\left[\mathbf{x}\mathbf{x}^T\right]$ where expectation is taken over activation vectors. This matrix is symmetric positive semi-definite having a *real* eigenvalue decomposition (EVD) $\mathbf{C}_{\mathbf{X}} = \mathbf{V}\mathbf{D}\mathbf{V}^T$ where $\mathbf{V}$ is an orthonormal matrix whose columns are eigenvectors, and $\mathbf{D}$ is a diagonal matrix containing the corresponding *non-negative* eigenvalues, assumed to be sorted in decreasing order. The eigenvector corresponding to the $i^{\text{th}}$ largest eigenvalue is called $i^{\text{th}}$ *principal* eigenvector.

This autocorrelation matrix can be empirically estimated using an instance of the activation tensor:

$$\mathbf{X} = [\mathbf{x}_1|\ldots|\mathbf{x_M}] \Rightarrow \mathbf{X}\mathbf{X}^T = \left[\mathbf{x}_1\mathbf{x}_1^T + \ldots + \mathbf{x}_M\mathbf{x}_M^T\right] \Rightarrow \mathbf{C}_{\mathbf{X}} = {}^{\mathbf{X}\mathbf{X}^T}/M \tag{4}$$

However, evaluating (4) and its EVD dynamically introduces a prohibitive computational overhead. Thus, we estimate $\mathbf{C}_{\mathbf{X}}$ in a pre-deployment calibration process. Specifically, during calibration, we sample and forward pass $B$ random input batches, and for each, calculate $\mathbf{C}_{\mathbf{X}}^{(i)} = \mathbf{X}^{(i)}\mathbf{X}^{(i)^T}/M$, where superscript $i$ denotes batch index. Then, we average our estimate of the auto-correlation matrix as $\mathbf{C}_{\mathbf{X}} = \sum_{i=1}^{B} \mathbf{C}_{\mathbf{X}}^{(i)}/B$ and use its eigenvalue decomposition for further optimizations.

This ergodic approach of estimating activation statistics as part of a calibration process has been employed to great effect in other compression works on quantization [26, 27] and pruning [28].

### 3.2 Activation decomposition with minimum mean squared error

Let us write $\tilde{\mathbf{X}} = \mathbf{P}\mathbf{P}^T\mathbf{X}$; for a vector $\mathbf{x} \in \mathbf{X}$, its counterpart in $\tilde{\mathbf{x}} \in \tilde{\mathbf{X}}$ is given by:

$$\tilde{\mathbf{x}} = \sum_{i=1}^{L}\langle\mathbf{p}_i, \mathbf{x}\rangle\mathbf{p}_i \tag{5}$$

where $\{\mathbf{p}_i\}_{i=1}^{L}$ are the orthonormal column vectors of $\mathbf{P}$, i.e., $\langle\mathbf{p}_i, \mathbf{p}_j\rangle = \mathbb{1}_{\{i==j\}}, \forall i, j \in 1\ldots L$.

We define the mean squared error (MSE) of the decomposition as

$$\mathbb{E}\left[\|\mathbf{x} - \tilde{\mathbf{x}}\|^2\right] \tag{6}$$

with the $L_2$-norm used throughout this paper. Our first result constructs $\mathbf{P}$ minimizing this MSE.

**Theorem 1.** *For an activation tensor $\mathbf{X}$ whose auto-correlation matrix has an eigenvalue decomposition given by $\mathbf{C_X} = \mathbf{VDV}^T$, the projection matrix $\mathbf{P}$ minimizing the mean squared error in (6) is given by $\mathbf{P} = [\mathbf{v}_1 | \dots | \mathbf{v}_L]$ where $\mathbf{v}_i$ is the $i^{th}$ principal eigenvector in $\mathbf{V}$.*

*Proof.* See Appendix A.1. The result is readily obtained by substituting $\tilde{\mathbf{x}}$ in (5) into (6) and minimizing the MSE which involves quadratic forms involving the positive semi-definite $\mathbf{C_X}$. $\quad\square$

Theorem 1 shares similarities with the Principal Component Analysis (PCA) algorithm [29]. PCA extracts low dimensional features having maximum correlation with input data. Unlike PCA, we omit input normalization to elide its computational cost. Still, we term the columns of $\mathbf{P}$ in Theorem 1 as Principal Activation Components. Since those are obtained using an EVD on a static estimation of $\mathbf{C_X}$, we call our method Eigen Static Principal Activation Component Estimation (ESPACE).

The MSE in Theorem 1 is a strong indicator of the quality of an approximation technique, e.g., it is often employed in quantization studies [26, 27]. However, empirical data may contain large outliers which can dominate the optimization process; say a few high-magnitude vectors in (6) masking the contribution of small data on the solution. An alternate metric to the MSE can be employed to prevent such artifacts in averaging: the normalized MSE (NMSE) defined as:

$$\mathbb{E}\left[\|\mathbf{x} - \tilde{\mathbf{x}}\|^2 / \|\mathbf{x}\|^2\right] \tag{7}$$

The solution of Theorem 1 can be slightly modified to minimize the NMSE in (7).

**Corollary 2.** *For an activation tensor $\mathbf{X}$, let $\hat{\mathbf{C}}_\mathbf{X} = \mathbb{E}\left[(\mathbf{x}/\|\mathbf{x}\|)(\mathbf{x}/\|\mathbf{x}\|)^T\right]$ be its input-normalized auto-correlation matrix having an eigenvalue decomposition given by $\hat{\mathbf{C}}_\mathbf{X} = \mathbf{VDV}^T$, the projection matrix $\mathbf{P}$ minimizing the normalized mean squared error in (7) is given by $\mathbf{P} = [\mathbf{v}_1 | \dots | \mathbf{v}_L]$ where $\mathbf{v}_i$ is the $i^{th}$ principal eigenvector in $\mathbf{V}$.*

*Proof.* The proof in Appendix A.2 uses equivalence of NMSE and MSE with $L_2$-normalized vectors. $\quad\square$

Corollary 2 applies to the decomposition in (3) with no activation normalization required at compute time. Rather, normalization is done during calibration, where $\hat{\mathbf{C}}_\mathbf{X}$ is estimated instead of $\mathbf{C_X}$.

Both solutions in Theorem 1 and Corollary 2 are options to be employed in ESPACE, where either may be more suitable on a layer-wise basis. Next we present further options for ESPACE based on the optimization of alternate metrics to the MSE and NMSE.

### 3.3 Activation decomposition with optimized forward propagated accuracy metrics

While local fidelity metrics, such as the MSE and NMSE above, are good indicators of the quality of an approximation technique, it has been shown that better insights on a neural network's accuracy may be derived via the study of forward propagated noise [30, 31, 32]. In this section, we study the effects of the decomposition in (3) on the output of the GEMM, and the output loss of the model.

At a given layer, let us write an arbitrary scalar in the GEMM output tensor in (1) as $y \in \mathbf{Y}$. Note that $y = \langle \mathbf{w}, \mathbf{x} \rangle$ for some weight vector in $\mathbf{w} \in \mathbf{W}$ and activation vector $\mathbf{x} \in \mathbf{X}$. We also let $\tilde{y}$ be the associated output when the GEMM is approximated by (3), which is given by $\tilde{y} = \langle \mathbf{w}, \tilde{\mathbf{x}} \rangle$ with $\tilde{\mathbf{x}}$ given by (5). We define the GEMM Output-referred MSE (GO-MSE) as $\mathbb{E}\left[(y - \tilde{y})^2\right]$.

Similarly, given an input to the network, we write the output loss function (the vocab cross-entropy) as $\mathcal{L}$. When one arbitrary activation tensor is transformed as per (3), a mismatch in computation is introduced and propagated all the way to the output. We let $\tilde{\mathcal{L}}$ be the resulting new value of the loss function. We define the Network Loss-referred MSE (NL-MSE) as $\mathbb{E}\left[(\mathcal{L} - \tilde{\mathcal{L}})^2\right]$.

A closed form solution for $\mathbf{P}$ in (3) minimizing the GO-MSE and NL-MSE is elusive to us. Therefore, we derive upper bounds on these metrics which we use as a proxy for optimization.

**Proposition 3.** *For a GEMM in (1) and its decomposition in (3), the GO-MSE is upper bounded by:*

$$\mathbb{E}\left[(y - \tilde{y})^2\right] \leq 2\mathbb{E}\left[\|\mathbf{w}\|^2 \cdot \|\mathbf{x}\|^2\right] - 2\mathbb{E}\left[\langle \mathbf{w}, \mathbf{x} \rangle \cdot \langle \mathbf{w}, \tilde{\mathbf{x}} \rangle\right] \tag{8}$$

*and the NL-MSE is upper bounded by*

$$\mathbb{E}\left[(\mathcal{L} - \tilde{\mathcal{L}})^2\right] \leq 2\mathbb{E}\left[\|\nabla_{\mathbf{x}}\|^2 \cdot \|\mathbf{x}\|^2\right] - 2\mathbb{E}\left[\langle\nabla_{\mathbf{x}}, \mathbf{x}\rangle \cdot \langle\nabla_{\mathbf{x}}, \tilde{\mathbf{x}}\rangle\right] \tag{9}$$

*where a first order Taylor approximation on the loss function is assumed and its gradient with respect to vector $\mathbf{x}$ is denoted as $\nabla_{\mathbf{x}}$.*

*Proof.* The proof in Appendix A.3 first shows $\|\tilde{\mathbf{x}}\|^2 < \|\mathbf{x}\|^2$ and then uses the Cauchy Schwarz inequality to establish both bounds. $\square$

Next, we provide closed form solutions for $\mathbf{P}$ in (3) minimizing the bounds in Proposition 3.

**Theorem 4.** *For a GEMM in (1) and its decomposition in (3), the projection matrix minimizing the bounds in Proposition 3 is given by $\mathbf{P} = [\mathbf{v}_1 | \ldots | \mathbf{v}_L]$ where $\mathbf{v}_i$ is the $i^{th}$ principal eigenvector in $\mathbf{V}$ obtained via eigenvalue decomposition on a matrix $\mathbf{C} = \mathbf{V}\mathbf{D}\mathbf{V}^T$ defined as:*

$$\mathbf{C} = \mathbb{E}\left[\mathbf{x}\mathbf{x}^T\mathbf{w}\mathbf{w}^T + \mathbf{w}\mathbf{w}^T\mathbf{x}\mathbf{x}^T\right] \quad and \quad \mathbf{C} = \mathbb{E}\left[\mathbf{x}\mathbf{x}^T\nabla_{\mathbf{x}}\nabla_{\mathbf{x}}^T + \nabla_{\mathbf{x}}\nabla_{\mathbf{x}}^T\mathbf{x}\mathbf{x}^T\right] \tag{10}$$

*to minimize the upper bounds on GO-MSE in (8) and NL-MSE in (9), respectively.*

*Proof.* The proof is included in Appendix A.4, where we also include modifications required in calibration. Specifically, $\mathbf{C_X}$ is reused and left/right multiplied by $\mathbf{w}\mathbf{w}^T/N$ to yield $\mathbf{C}$ in (10) minimizing the bound on GO-MSE. An additional backward pass is needed to properly scale activation vectors and their gradients when calibrating $\mathbf{C}$ in (10) minimizing the bound on NL-MSE. $\square$

Theorem 4 augments Theorem 1 and Corollary 2 with two options for the design of $\mathbf{P}$. Much like Corollary 2, we supplement our new solutions with $L_2$-normalization to include

$$\hat{\mathbf{C}} = \mathbb{E}\left[\left(\mathbf{x}\mathbf{x}^T\mathbf{w}\mathbf{w}^T + \mathbf{w}\mathbf{w}^T\mathbf{x}\mathbf{x}^T\right)/\|\mathbf{w}\|^2 \cdot \|\mathbf{x}\|^2\right] \quad and \quad \hat{\mathbf{C}} = \mathbb{E}\left[\left(\mathbf{x}\mathbf{x}^T\nabla_{\mathbf{x}}\nabla_{\mathbf{x}}^T + \nabla_{\mathbf{x}}\nabla_{\mathbf{x}}^T\mathbf{x}\mathbf{x}^T\right)/\|\nabla_{\mathbf{x}}\|^2 \cdot \|\mathbf{x}\|^2\right]$$

as alternate choices for the calibrated matrices $\mathbf{C}$ in (10). Unlike Corollary 2, $L_2$-normalization in these two matrices does not correspond to a notable optimization. Nevertheless, these options are retained in the spirit of suppressing the influence of large data in calibration.

Thus, overall we have six choices for $\mathbf{P}$. Since each can be obtained as part of a fast and pre-deployment calibration phase, we may simply select the best one for each layer. In our experiments of Section 4, the best candidate is determined via a per-layer validation over all six choices. A sensitivity study on the impact of each of the six candidates is provided in Appendix B.4.

## 4 Model Compression Studies

In this section, we report on experimental studies investigating LLM compression using ESPACE.

### 4.1 Experimental setup

We employ three sets of open source LLMs: GPT3 [33], Llama2 [34], and Nemotron4 [35]. Specifically, we experiment on GPT3-{1.3B, 8B, 22B}, Llama2-{7B, 13B}, and Nemotron4-15B. Accuracy is evaluated in two ways: perplexity measured on the Wikitext-103 dataset [36] and zero-shot downstream task accuracy of: BoolQ (BQ) [37], Hellaswag (HS) [38], PIQA (PQ) [39], RACE (RA) [40], and WinoGrande (WG) [41].

The Wikitext-103 dataset is split into train, validation, and test sets. We use 512 random sequences from the training set for calibrating projection matrices required by ESPACE. We use the validation set for layer-wise sensitivity studies. The test set is used to report perplexity results in this section.

Our implementation uses NVIDIA's Megatron LM [33] and downstream task evaluation invokes Eleuther AI's LM evaluation harness [42]. For the latter, we report raw accuracy scores, and their average; we do not post process results or apply normalization to the scores.

When ESPACE is applied, we retrain the models to adapt to the approximation error of activation projection as discussed in Section 2. Retraining simply extends the models' pre-training sessions and

uses the 330B-token MTNLG dataset [43], which was used to train GPT3 models. All implementation details are included in Appendix B to help reproducibility of our results.

We metricize model size reduction via inference compression rate. Specifically, for layers decomposed per (3), we count the number of entries in $\mathbf{P}$ and $\left(\mathbf{P}^T\mathbf{W}\right)$; for other layers, we count those in $\mathbf{W}^T$. We also report the latency of executing all network GEMMs in (1) or (3), which we measure using a NVIDIA A100 GPU and a simple, un-optimized implementation (see Appendix B.4). We also report prefill inference latency, metricized via the Time to First Token (TTFT), and measured using the Megatron-LM implementation. In our measurements, we use a batch size of 1 and sequence length of 2048 and 4096 for GPT3 and Llama2/Nemotron4 models, respectively. The reported reductions in total GEMM latency and TTFT constitute evidence that compression improves inference throughput. It is beyond the scope of this paper to evaluate the impact of ESPACE on end-to-end token throughput and latency on LLM inference serving systems, since this requires a complex set of optimizations including but not limited to optimization of back-to-back GEMMs into fused kernels, KV caching, continuous batching, as well as thorough performance studies with varying input and output sequence lengths. Thus, we leave an evaluation of token generation throughput and energy savings and improvements to future work.

## 4.2 Validation perplexity studies

Our experiments start with a calibration phase where we prepare the static projection matrix $\mathbf{P}$ for each layer. The dimension $L$ in $\mathbf{P}$ is chosen as the lowest power of two such that layer compression is at least 50%. The power-of-two restriction ensures best tensor core utilization, and the resulting compression rate depends on the dimensions of the original layer ($K$ and $N$). Exact details of these values for all layers and models are included in Appendix B.2.

We perform a sensitivity study on the Wikitext-103 validation perplexity when ESPACE is applied out-of-the-box (no retraining) one layer at a time. For each layer, we identify which of our six candidates projection matrices in Section 3 yields lowest validation perplexity. Layers are then sorted according to their impact on perplexity from least to most destructive. We then evaluate the validation perplexity when ESPACE is progressively applied to out-of-the-box to all layers according to this ranking. Fine-grained details of this exploration are included in Appendix C for all models.

This exploration yields an interesting finding: as we progressively apply ESPACE to more layers, the perplexity marginally increases until an inflection point after which accuracy degradation accelerates. This inflection occurs at 20% to 40% compression depending on the model. Figure 3 depicts this phenomenon for GPT3-22B, and the same data for other models can be found in Appendix C.2.

We find that out-of-the-box application of ESPACE works better for larger models; GPT3-22B, the largest model we experimented on, exhibits an inflection in perplexity at 40% compression, which is the highest in our results. This is consistent with many earlier works on general compression of neural networks [44, 45, 46, 47]. Interestingly, a 20% out-of-the-box compressed GPT3-22B is iso-accurate to its uncompressed counterpart (see Figure 3); without retraining, its test perplexity of **6.61** which is within 1% of the 6.55 baseline.

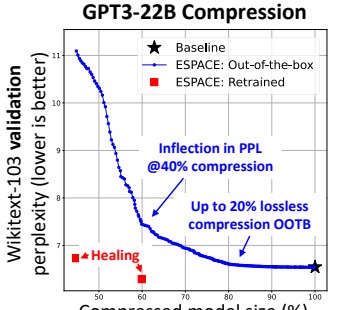

Figure 3: Validation perplexity for GPT3-22B when ESPACE is progressively applied to its GEMM layers. The order of layer selection is based on a layer-wise sensitivity analysis.

After the above validation study is performed, we select two configurations for layers to be compressed using ESPACE: (a) layers corresponding to the inflection point, i.e., 20% to 40% compression, and (b) as many layers needed to achieve a compression of ∼50%. For both configurations, we retrain the compressed models and further evaluate their achievable accuracy.

## 4.3 Compression of GPT3 models

Once compression targets and layer configurations are set, we retrain GPT3 models on the MTNLG dataset. Although we use all of the 330B available tokens, we do observe the training loss quickly

Table 1: GEMM latency, time to first token, Wikitext-103 perplexity (WK-103 PPL), and downstream task accuracy of GPT3, Llama2, and Nemotron4 models compressed with ESPACE[3].

| Method (Compression) | # of Weights | Total GEMM Latency | TTFT impl. in Megatron-LM | WK-103 PPL ↓ | Downstream Task Accuracy ↑ | | | | | |
|---|---|---|---|---|---|---|---|---|---|---|
| | | | | | BQ | HS | PQ | RA | WG | Avg. |
| GPT3-1.3B | | | | | | | | | | |
| Baseline | $1.21 \times 10^9$ | 24.2ms | 39.8ms | 9.94 | **64.3** | 43.5 | **74.2** | **37.6** | 58.1 | 55.5 |
| ESPACE (20%) | $9.71 \times 10^8$ | 20.6ms (-15%) | 36.1ms (-9%) | **9.53** | 60.6 | **45.1** | *73.0* | *36.4* | **62.9** | **55.6** |
| ESPACE (47%) | $6.42 \times 10^8$ | 15.9ms (-34%) | 31.7ms (-20%) | 11.07 | *62.3* | 39.9 | *71.6* | 34.5 | 58.7 | 53.4 |
| GPT3-8B | | | | | | | | | | |
| Baseline | $8.05 \times 10^9$ | 136ms | 186ms | 7.38 | 69.0 | 54.2 | 78.1 | **41.4** | 67.8 | 62.1 |
| ESPACE (21%) | $6.33 \times 10^9$ | 110ms (-19%) | 155ms (-16%) | **7.00** | **70.3** | **55.3** | **78.9** | 40.7 | **69.3** | **62.9** |
| ESPACE (50%) | $4.08 \times 10^9$ | 76.8ms (-44%) | 122ms (-35%) | *7.66* | *66.5* | *52.3* | *77.6* | 38.9 | *66.9* | *60.4* |
| GPT3-22B | | | | | | | | | | |
| Baseline | $2.17 \times 10^{10}$ | 354ms | 457ms | 6.55 | 76.4 | 57.2 | 79.3 | **40.7** | 70.5 | 64.8 |
| ESPACE (40%) | $1.30 \times 10^{10}$ | 229ms (-35%) | 313ms (-32%) | **6.29** | **76.6** | **57.3** | **79.5** | *40.2* | *70.2* | *64.8* |
| ESPACE (55%) | $9.74 \times 10^9$ | 181ms (-49%) | 261ms(-43%) | *6.73* | *72.2* | *55.8* | *79.3* | *40.1* | *69.7* | *63.4* |
| Llama2-7B | | | | | | | | | | |
| Baseline | $6.48 \times 10^9$ | 210ms | 368ms | **5.06** | **79.2** | 57.1 | 78.1 | **44.0** | 69.5 | 65.6 |
| Retrained (0%) | $6.48 \times 10^9$ | 210ms | 368ms | **5.06** | 78.2 | **57.9** | 78.0 | 43.7 | **70.6** | **65.7** |
| ESPACE (21%) | $5.11 \times 10^9$ | 169ms (-19%) | 322ms (-12%) | *5.07* | *77.1* | *57.1* | *78.7* | *42.7* | *69.2* | *65.0* |
| ESPACE (50%) | $3.24 \times 10^9$ | 113ms (-46%) | 266ms (-28%) | 5.67 | 72.2 | 52.0 | 76.5 | 38 | 63.5 | 60.4 |
| Llama2-13B | | | | | | | | | | |
| Baseline | $1.27 \times 10^{10}$ | 406ms | 643ms | 4.61 | **82.4** | 60.2 | **79.5** | **46.8** | 71.9 | **68.2** |
| ESPACE (20%) | $1.01 \times 10^{10}$ | 336ms (-17%) | 562ms (-13%) | **4.59** | *78.3* | **60.5** | **79.5** | 43.0 | **72.8** | 66.8 |
| ESPACE (50%) | $6.34 \times 10^9$ | 259ms (-36%) | 447ms (-31%) | 5.13 | 75.7 | 56.2 | 78.0 | 41.5 | *69.1* | 64.1 |
| Nemotron4-15B | | | | | | | | | | |
| Baseline | $1.25 \times 10^{10}$ | 414ms | 741ms | **6.06** | 78.3 | **62.1** | **81.1** | **47.0** | **75.2** | **68.7** |
| ESPACE (25%) | $9.54 \times 10^9$ | 324ms (-22%) | 655ms (-12%) | *6.28* | **78.9** | 59.9 | *80.0* | *46.4* | *72.8* | *67.6* |
| ESPACE (50%) | $6.25 \times 10^9$ | 223ms (-46%) | 545ms (-26%) | 6.93 | *77.9* | 57.0 | *77.8* | 42.4 | *69.9* | 65.0 |

converging. We leave training hyperparameters unchanged except for one: we disable dropout. Our rationale is that activation projection is one form of deterministic and structured dropout such that additional regularization may not be needed. Results[3] on GPT3 models are included in Table 1.

We find that ESPACE can compress GPT3 models by ∼50% at the cost of a small accuracy degradation. In the case of GPT3-22B, the perplexity increase is of only 0.18; in general, the gap decreases for larger overall model size. By and large, similar trends are observed for downstream task accuracies and we note that most scores of 50% compressed models fall within 5% of the baseline.

For lower compression ratios (inflection points at 20% to 40%), ESPACE converges to an accuracy *better than that of the baseline*. The best improvement occurs for GPT3-8B, where ESPACE produces a 6B model with 0.38 lower perplexity than its 8B baseline. The improvements are observed both in terms of perplexity and downstream task accuracy. While GPT3 models may be over-parameterized, we posit that ESPACE acts a regularizer at moderate compression rates. Specifically, we believe that projection onto principal activation components filters out unnecessary information coming from small eigenvalue components.

For all models, we observe an encouraging translation of compression to GEMM latency reduction by up to 49% which leads to noticeable speed-up in TTFT by up to 43%..

## 4.4 Compression of Llama2 models and comparison to related works

For Llama2, we only retrain using 200B MTNLG tokens because we observed quick convergence for GPT3. Llama2 models were trained on an undisclosed dataset of 2T tokens [34]. Therefore, with 200B tokens, the healing phase of ESPACE constitutes no more than 10% of the original pre-training session. Since Llama2 pre-training details are not openly available, we re-used all hyperparameters from GPT3, which is likely to be sub-optimal. In spite of the two handicaps of dataset disparity and hyperparameter sub-optimality, we obtained promising results as reported in Table 1.

For Llama2-7B, we first retrained the uncompressed baseline. The purpose of this experiment is twofold: (a) ensure that our hyperparameters at least do not corrupt the model, and (b) verify that the

---

[3]**Boldface** indicates best result per task. *Italics* indicates ESPACE results within 5% of the baseline

healing process is not just an artifact of processing more tokens. Both hypotheses appeared to be valid: the retrained baseline has nearly identical accuracy compared to the original model.

Generally, we find that the trends of ESPACE compression for Llama2 are similar to GPT3, albeit slightly less successful. Though the results are still promising, we attribute the slight shortcomings in accuracy to the handicaps above. We find that 50% ESPACE compression on Llama2 leads to ∼0.6 perplexity increase and similar degradation in terms of downstream task accuracy. Notably, compressing the Llama2-13B model to to a 6.3B model yields comparable accuracy to the Llama2-7B baseline which itself is a 6.5B model.

In addition, for 20% compression, we find that ESPACE matches the accuracy of the baseline for Llama2 models. While not as impressive as the improvements observed with GPT3, ESPACE is able to produce 5B and 10B models matching the 7B and 13B baselines, which does push the pareto frontier of accuracy versus model size in the right direction as shown in Figure 1.

The Llama2-7B model has been used in related works on tensor decomposition mentioned in Section 1.1; specifically, ASVD [20], SVD-LoRa [18], and SliceGPT [21]. Both ASVD and sliceGPT have reported perplexity on Wikitext, but SVD-LoRa performed task-specific finetuning on a variety of datasets and averaged perplexities. Therefore, in Figure 4, we compare our results to these works using perplexity increase over baseline, rather than raw perplexity, for maximum inclusivity.

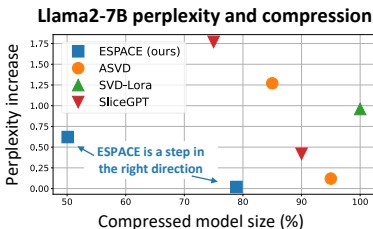

Figure 4: Comparison to related works compressing Llama2-7B using matrix factorization techniques.

SVD-LoRa performed an SVD decomposition on the weights such that the intermediate dimension is half of dot-product which leads to no compression. On the other hand, ASVD and sliceGPT can only achieve modest compression ratios of up to 25% with some loss in accuracy. Recall that these works apply factorization on weights which is the fundamental difference to ESPACE. As seen in Figure 4, ESPACE is a step in the right direction towards improving the state-of-the-art in tensor decomposition of LLMs.

## 4.5 Compression of Nemotron4-15B

Finally, we used ESPACE to compress Nemotron4-15B into 9.54 and 6.25 billion parameters, as reported in Table 1. Retraining consumed 275B tokens which corresponds to ∼ 3% of this model's original training session. Once more, compression with ESPACE leads to minimal degradation in the moderate regime (25%) and yields a small accuracy drop in the aggressive regime (50%).

Consistently with our findings for the above models, ESPACE reduces GEMM execution time by up to 46%. This, in turn, improves the TTFT by up to 26%. An interesting observation is that, for Llama2 and Nemotron4 models, the TTFT improvement is slightly less pronounced than for GPT3 models. This is simply due to the fact that the latter uses a sequence length of 2048, whereas the former two use 4096. A larger sequence length means more time is spent in attention cross-activation products which amortizes the speed-up in the GEMM layers.

## 5   Conclusion

We have presented ESPACE, a novel compression technique realizing tensor decomposition of LLMs in an activation-centric manner. A set of theoretical results were derived to guide the construction of activation projection which is done statically. Experimentally, we have shown promising results where ESPACE is able to ∼50% compress modern LLMs at the cost of a small accuracy degradation. Compared to related works, ESPACE is a first step in pushing the frontier of model size versus accuracy trade-offs. Future work includes combining ESPACE with alternate compression techniques such as quantization and pruning, and evaluating decomposition of activation tensors in attention. As potential extension to our algorithm, the use of matrix sketching and random projections may pave the way for better overall compressibility.

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

# A  Proofs of theoretical results

In this first appendix, we provide proofs for the various theoretical results in Section 3.

## A.1  Proof of Theorem 1

For a pair of vectors $\mathbf{x} \in \mathbf{X}$ and $\tilde{\mathbf{x}} \in \tilde{\mathbf{X}}$, and using (5), we have the squared error:

$$\|\mathbf{x} - \tilde{\mathbf{x}}\|^2 = \|\mathbf{x}\|^2 + \|\tilde{\mathbf{x}}\|^2 - 2\mathbf{x}^T\tilde{\mathbf{x}} = \|\mathbf{x}\|^2 + \|\tilde{\mathbf{x}}\|^2 - 2\sum_{i=1}^{L} \left(\mathbf{p_i}^T\mathbf{x}\right)\mathbf{p_i}^T\mathbf{x}$$

$$\Rightarrow \|\mathbf{x} - \tilde{\mathbf{x}}\|^2 = \|\mathbf{x}\|^2 + \|\tilde{\mathbf{x}}\|^2 - 2\sum_{i=1}^{L} \left(\mathbf{p_i}^T\mathbf{x}\right)^2$$

Furthermore, note that

$$\|\tilde{\mathbf{x}}\|^2 = \tilde{\mathbf{x}}^T\tilde{\mathbf{x}} = \left(\sum_{i=1}^{L}\left(\mathbf{p_i}^T\mathbf{x}\right)\mathbf{p_i}\right)^T \left(\sum_{i=1}^{L}\left(\mathbf{p_i}^T\mathbf{x}\right)\mathbf{p_i}\right)$$

and since $\{\mathbf{p}_i\}_{i=1}^{L}$ are orthonromal, cross product terms vanish and we have: $\|\tilde{\mathbf{x}}\|^2 = \sum_{i=1}^{L}\left(\mathbf{p_i}^T\mathbf{x}\right)^2$ which we plug back into the expression for the squared error:

$$\|\mathbf{x} - \tilde{\mathbf{x}}\|^2 = \|\mathbf{x}\|^2 + \sum_{i=1}^{L}\left(\mathbf{p_i}^T\mathbf{x}\right)^2 - 2\sum_{i=1}^{L}\left(\mathbf{p_i}^T\mathbf{x}\right)^2 = \|\mathbf{x}\|^2 - \sum_{i=1}^{L}\left(\mathbf{p_i}^T\mathbf{x}\right)^2$$

Furthermore, note that:

$$\left(\mathbf{p_i}^T\mathbf{x}\right)^2 = \left(\mathbf{p_i}^T\mathbf{x}\right)\left(\mathbf{p_i}^T\mathbf{x}\right) = \left(\mathbf{p_i}^T\mathbf{x}\right)\left(\mathbf{x}^T\mathbf{p_i}\right) = \mathbf{p_i}^T\left(\mathbf{x}\mathbf{x}^T\right)\mathbf{p_i}$$

where we used the commutativity of dot product and associativity of matrix multiplication. Thus the squared error is given by:

$$\|\mathbf{x} - \tilde{\mathbf{x}}\|^2 = \|\mathbf{x}\|^2 - \sum_{i=1}^{L}\mathbf{p_i}^T\left(\mathbf{x}\mathbf{x}^T\right)\mathbf{p_i}$$

Finally, we take expectation on both sides and obtain a formula for the MSE:

$$\mathbb{E}\left[\|\mathbf{x} - \tilde{\mathbf{x}}\|^2\right] = \mathbb{E}\left[\|\mathbf{x}\|^2\right] - \mathbb{E}\left[\sum_{i=1}^{L}\mathbf{p_i}^T\left(\mathbf{x}\mathbf{x}^T\right)\mathbf{p_i}\right] = \mathbb{E}\left[\|\mathbf{x}\|^2\right] - \sum_{i=1}^{L}\mathbf{p_i}^T\mathbb{E}\left[\mathbf{x}\mathbf{x}^T\right]\mathbf{p_i}$$

where we used linearity of expectation and the fact that $\{\mathbf{p}_i\}_{i=1}^{L}$ are not random. In this formula for the MSE, $\mathbb{E}\left[\|\mathbf{x}\|^2\right]$ does not depend on $\{\mathbf{p}_i\}_{i=1}^{L}$, and therefore, minimizing the MSE is equivalent to **maximizing** the following expression involving the auto-correlation matrix:

$$\sum_{i=1}^{L}\mathbf{p_i}^T\mathbb{E}\left[\mathbf{x}\mathbf{x}^T\right]\mathbf{p_i} = \sum_{i=1}^{L}\mathbf{p_i}^T\mathbf{C_X}\mathbf{p_i}$$

where each term in the summation is a quadratic form on the positive semi-definite auto-correlation matrix $\mathbf{C_X}$. Since $\{\mathbf{p}_i\}_{i=1}^{L}$ are orthonormal, this is an equivalent form of the Rayleigh quotient [48] and the solution is to assign $\{\mathbf{p}_i\}_{i=1}^{L}$ as the $L$ principal eigenvectors of $\mathbf{C_X}$. This concludes the proof of Theorem 1.

## A.2 Proof of Corollary 2

The result can be readily obtained as a consequence of the following:

$$\mathbb{E}\left[\frac{\|\tilde{\mathbf{x}} - \mathbf{x}\|^2}{\|\mathbf{x}\|^2}\right] = \mathbb{E}\left[\left\|\frac{\tilde{\mathbf{x}}}{\|\mathbf{x}\|} - \frac{\mathbf{x}}{\|\mathbf{x}\|}\right\|^2\right] = \mathbb{E}_{\mathbf{X}}\left[\left\|\frac{\sum_{i=1}^{L}\langle\mathbf{p}_i,\mathbf{x}\rangle\mathbf{p}_i}{\|\mathbf{x}\|} - \frac{\mathbf{x}}{\|\mathbf{x}\|}\right\|^2\right]$$

$$\Rightarrow \mathbb{E}\left[\frac{\|\tilde{\mathbf{x}} - \mathbf{x}\|^2}{\|\mathbf{x}\|^2}\right] = \mathbb{E}\left[\left\|\sum_{i=1}^{L}\langle\mathbf{p}_i,\frac{\mathbf{x}}{\|\mathbf{x}\|}\rangle\mathbf{p}_i - \frac{\mathbf{x}}{\|\mathbf{x}\|}\right\|^2\right]$$

Therefore, the setup is identical to that of Theorem 1 and we may apply the same solution as Appendix A.1 above. The only difference is that activation vectors are $L_2$-normalized which is why $\hat{\mathbf{C}}_{\mathbf{X}}$ (which is also positive semi-definite) is used in lieu of $\mathbf{C}_{\mathbf{X}}$ in Corollary 2.

## A.3 Proof of Proposition 3

A preliminary result needed is to show that for any activation vector, we have $\|\tilde{\mathbf{x}}\|^2 < \|\mathbf{x}\|^2$. We first note that while $\tilde{\mathbf{x}} = \sum_{i=1}^{L}\langle\mathbf{p}_i,\mathbf{x}\rangle\mathbf{p}_i$ per (5), we also have that $\mathbf{x} = \sum_{i=1}^{K}\langle\mathbf{p}_i,\mathbf{x}\rangle\mathbf{p}_i$, where $\{\mathbf{p}_i\}_{i=1}^{K}$ extend the set of orthonormal vectors $\{\mathbf{p}_i\}_{i=1}^{L}$ to be complete, i.e., equivalent to no truncation of columns of the full rank matrix $\mathbf{V}$ when constructing $\mathbf{P}$, regardless of the metric being optimized.

Using orthonormality of projection vectors, similar to the proof of Theorem 1 in Appendix A.1 above, we obtain $\|\tilde{\mathbf{x}}\|^2 = \sum_{i=1}^{L}\left(\mathbf{p_i}^T\mathbf{x}\right)^2$ and $\|\mathbf{x}\|^2 = \sum_{i=1}^{K}\left(\mathbf{p_i}^T\mathbf{x}\right)^2$. Therefore:

$$\|\mathbf{x}\|^2 - \|\tilde{\mathbf{x}}\|^2 = \sum_{i=L+1}^{K}\left(\mathbf{p_i}^T\mathbf{x}\right)^2 \geq 0 \Rightarrow \|\tilde{\mathbf{x}}\|^2 < \|\mathbf{x}\|^2$$

where we used the fact that a sum of non-negative quantities is non-negative.

Then for a scalar $y \in \mathbf{Y}$ and its counterpart $\tilde{y} \in \tilde{\mathbf{Y}}$, we have:

$$\begin{aligned}
(y - \tilde{y})^2 = \left(\mathbf{w}^T\mathbf{x} - \mathbf{w}^T\tilde{\mathbf{x}}\right)^2 &= \left(\mathbf{w}^T\mathbf{x}\right)^2 + \left(\mathbf{w}^T\tilde{\mathbf{x}}\right)^2 - 2\left(\mathbf{w}^T\mathbf{x}\mathbf{w}^T\tilde{\mathbf{x}}\right) \\
&\leq \|\mathbf{w}\|^2 \cdot \|\mathbf{x}\|^2 + \|\mathbf{w}\|^2 \cdot \|\tilde{\mathbf{x}}\|^2 - 2\left(\mathbf{w}^T\mathbf{x}\mathbf{w}^T\tilde{\mathbf{x}}\right) \\
&\leq \|\mathbf{w}\|^2 \cdot \|\mathbf{x}\|^2 + \|\mathbf{w}\|^2 \cdot \|\mathbf{x}\|^2 - 2\left(\mathbf{w}^T\mathbf{x}\mathbf{w}^T\tilde{\mathbf{x}}\right) \\
&= 2\|\mathbf{w}\|^2 \cdot \|\mathbf{x}\|^2 - 2\left(\mathbf{w}^T\mathbf{x}\mathbf{w}^T\tilde{\mathbf{x}}\right)
\end{aligned}$$

where the first upper bound uses the Cauchy-Schwarz inequality while the second uses $\|\tilde{\mathbf{x}}\|^2 < \|\mathbf{x}\|^2$ which we proved above. Taking expectations on both sides of the inequality yields the upper bound on GO-MSE in (8).

Next, when a first order Taylor approximation on the loss function is assumed, we have the following relation between the unperturbed loss value $\mathcal{L}$ and its counterpart $\tilde{\mathcal{L}}$ when an activation vector $\mathbf{x}$ is projected to $\tilde{\mathbf{x}}$ per (5):

$$\tilde{\mathcal{L}} = \mathcal{L} + \nabla_{\mathbf{x}}^T(\tilde{\mathbf{x}} - \mathbf{x}) \Rightarrow \tilde{\mathcal{L}} - \mathcal{L} = \nabla_{\mathbf{x}}^T\tilde{\mathbf{x}} - \nabla_{\mathbf{x}}^T\mathbf{x}$$

$$\Rightarrow \left(\tilde{\mathcal{L}} - \mathcal{L}\right)^2 = \left(\nabla_{\mathbf{x}}^T\tilde{\mathbf{x}} - \nabla_{\mathbf{x}}^T\mathbf{x}\right)^2 = \left(\nabla_{\mathbf{x}}^T\tilde{\mathbf{x}}\right)^2 + \left(\nabla_{\mathbf{x}}^T\mathbf{x}\right)^2 - 2\left(\nabla_{\mathbf{x}}^T\mathbf{x}\nabla_{\mathbf{x}}^T\tilde{\mathbf{x}}\right)$$

once more, we use the Cauchy-Schwarz inequality and the fact that $\|\tilde{\mathbf{x}}\|^2 < \|\mathbf{x}\|^2$ to establish:

$$\left(\nabla_{\mathbf{x}}^T\mathbf{x}\right)^2 \leq \|\nabla_{\mathbf{x}}\|^2 \cdot \|\mathbf{x}\|^2 \quad \& \quad \left(\nabla_{\mathbf{x}}^T\tilde{\mathbf{x}}\right)^2 \leq \|\nabla_{\mathbf{x}}\|^2 \cdot \|\tilde{\mathbf{x}}\|^2 \leq \nabla_{\mathbf{x}}\|^2 \cdot \|\mathbf{x}\|^2$$

which we plug into the difference in network losses above to obtain:

$$\left(\tilde{\mathcal{L}} - \mathcal{L}\right)^2 \leq 2\|\nabla_{\mathbf{x}}\|^2 \cdot \|\mathbf{x}\|^2 - 2\left(\nabla_{\mathbf{x}}^T\mathbf{x}\nabla_{\mathbf{x}}^T\tilde{\mathbf{x}}\right)$$

Taking expectations on both sides yields the upper bound on NL-MSE in (9). This completes the proof of Proposition 3.

## A.4 Proof of Theorem 4

In order to minimize the upper bound on the GO-MSE in (8), note that it suffices to maximize the quantity $2\mathbb{E}\left[\langle \mathbf{w}, \mathbf{x}\rangle \cdot \langle \mathbf{w}, \tilde{\mathbf{x}}\rangle\right]$ since $2\mathbb{E}\left[\|\mathbf{w}\|^2 \cdot \|\mathbf{x}\|^2\right]$ does not depend on $\{\mathbf{p}_i\}_{i=1}^L$. We have the following:

$$2\langle \mathbf{w}, \mathbf{x}\rangle \cdot \langle \mathbf{w}, \tilde{\mathbf{x}}\rangle = 2\mathbf{w}^T\mathbf{x}\mathbf{w}^T\left(\sum_{i=1}^L \left(\mathbf{p}_i^T\mathbf{x}\right)\mathbf{p}_i\right) = 2\sum_{i=1}^L \mathbf{w}^T\mathbf{x}\mathbf{w}^T\mathbf{p}_i\mathbf{p}_i^T\mathbf{x}$$

$$= \sum_{i=1}^L \left(\mathbf{w}^T\mathbf{x}\mathbf{w}^T\mathbf{p}_i\mathbf{p}_i^T\mathbf{x} + \mathbf{w}^T\mathbf{x}\mathbf{w}^T\mathbf{p}_i\mathbf{p}_i^T\mathbf{x}\right)$$

But since the dot product is commutative, i.e., $\mathbf{a}^T\mathbf{b} = \mathbf{b}^T\mathbf{a}$ for any two vectors $\mathbf{a}$ and $\mathbf{b}$, we may rearrange each of the two identical terms inside the summation as follows:

$$\mathbf{w}^T\mathbf{x}\mathbf{w}^T\mathbf{p}_i\mathbf{p}_i^T\mathbf{x} = \mathbf{p}_i^T\mathbf{x}\mathbf{x}^T\mathbf{w}\mathbf{w}^T\mathbf{p}_i \quad \& \quad \mathbf{w}^T\mathbf{x}\mathbf{w}^T\mathbf{p}_i\mathbf{p}_i^T\mathbf{x} = \mathbf{p}_i^T\mathbf{w}\mathbf{w}^T\mathbf{x}\mathbf{x}^T\mathbf{p}_i$$

Therefore, we obtain:

$$2\langle \mathbf{w}, \mathbf{x}\rangle \cdot \langle \mathbf{w}, \tilde{\mathbf{x}}\rangle = \sum_{i=1}^L \left(\mathbf{p}_i^T\mathbf{x}\mathbf{x}^T\mathbf{w}\mathbf{w}^T\mathbf{p}_i + \mathbf{p}_i^T\mathbf{w}\mathbf{w}^T\mathbf{x}\mathbf{x}^T\mathbf{p}_i\right)$$

$$= \sum_{i=1}^L \mathbf{p}_i^T\left(\mathbf{x}\mathbf{x}^T\mathbf{w}\mathbf{w}^T + \mathbf{w}\mathbf{w}^T\mathbf{x}\mathbf{x}^T\right)\mathbf{p}_i$$

Taking expectations, we find that the quantity that needs to be maximized in order to minimize the bound on GO-MSE in (8) is:

$$\sum_{i=1}^L \mathbf{p}_i^T\mathbb{E}\left[\mathbf{x}\mathbf{x}^T\mathbf{w}\mathbf{w}^T + \mathbf{w}\mathbf{w}^T\mathbf{x}\mathbf{x}^T\right]\mathbf{p}_i$$

Similar to the proof of Theorem 1 in Appendix A.1, this is yet again a sum of a quadratic form over the orthonormal set of vectors $\{\mathbf{p}_i\}_{i=1}^L$ and the solution is therefore to assign these vectors as the $L$ principal vectors of $\mathbf{C} = \mathbb{E}\left[\mathbf{x}\mathbf{x}^T\mathbf{w}\mathbf{w}^T + \mathbf{w}\mathbf{w}^T\mathbf{x}\mathbf{x}^T\right]$ as per (10).

Note that the derivation above decomposed the dot products to obtain a quadratic form on the matrix $\mathbf{x}\mathbf{x}^T\mathbf{w}\mathbf{w}^T + \mathbf{w}\mathbf{w}^T\mathbf{x}\mathbf{x}^T$ because its symmetry is required for real eigenvalue decomposition. Also note that if positive definiteness is not achieved, we sort absolute values of eigenvalues. Finally, observe that this solution requires no overhead on the calibration process. Indeed, assuming weights and activations are independent, we note that $\mathbb{E}\left[\mathbf{x}\mathbf{x}^T\mathbf{w}\mathbf{w}^T + \mathbf{w}\mathbf{w}^T\mathbf{x}\mathbf{x}^T\right] = \mathbf{C_X}\mathbf{C_W} + \mathbf{C_W}\mathbf{C_X}$ where $\mathbf{C_W} = \mathbb{E}\left[\mathbf{w}\mathbf{w}^T\right]$ is the weight auto-correlation matrix, which can simply be calibrated as $\mathbf{C_W} = \mathbf{W}\mathbf{W}^T/N$. Thus we only require left and right scaling of the calibrated auto-correlation matrix.

Similarly, to minimize the upper bound on NL-MSE in (9), it suffices to maximize $2\mathbb{E}\left[\langle\nabla_\mathbf{x}, \mathbf{x}\rangle \cdot \langle\nabla_\mathbf{x}, \tilde{\mathbf{x}}\rangle\right]$. Using the exact same derivation as the above, replacing $\mathbf{w}$ by $\nabla_\mathbf{x}$, we obtain that the quantity to be maximized is

$$\sum_{i=1}^L \mathbf{p}_i^T\mathbb{E}\left[\mathbf{x}\mathbf{x}^T\nabla_\mathbf{x}\nabla_\mathbf{x}^T + \nabla_\mathbf{x}\nabla_\mathbf{x}^T\mathbf{x}\mathbf{x}^T\right]\mathbf{p}_i$$

which is done by assigning $\{\mathbf{p}_i\}_{i=1}^L$ as the $L$ principal vectors of $\mathbf{C} = \mathbb{E}\left[\mathbf{x}\mathbf{x}^T\nabla_\mathbf{x}\nabla_\mathbf{x}^T + \nabla_\mathbf{x}\nabla_\mathbf{x}^T\mathbf{x}\mathbf{x}^T\right]$ as per (10). Calibrating this matrix does require an extra step, where we perform a backward pass to estimate activation gradients. For ease of implementation, in our results, we make an approximation on the per-sequence independence of activation vectors and their gradients. This greatly reduces the memory requirements of the calibration process. And for each sample sequence in the calibration set, we compute $\mathbf{C}^{(i)} = \left(\mathbf{X}^{(i)}\mathbf{X}^{(i)T}\mathbf{G}^{(i)}\mathbf{G}^{(i)T} + \mathbf{G}^{(i)}\mathbf{G}^{(i)T}\mathbf{X}^{(i)}\mathbf{X}^{(i)T}\right)/M^2$ where $\mathbf{G}^{(i)}$ is the gradient tensor (whose vectors are instantiations of $\nabla_\mathbf{x}$). As always, we end the calibration phase by averaging across samples: $\mathbf{C} = \sum_{i=1}^B \mathbf{C}^{(i)}/B$. This completes the proof of Theorem 4.

# B    Implementation details

In this appendix, we discuss all details behind our implementation in Section 4. These details include per-model specific application of ESPACE as well as retraining recipes. We strive to provide excessive details such that independent reproducibility of our results is seamless. We also encourage readers to reach out to us (after blind reviewing) for any questions on implementations.

## B.1    Software implementation

As was mentioned in Section 4, our implementation is built on top of Megatron-LM [33] which itself is based on the Pytorch framework. We use Pytorch for all extra introductions needed by ESPACE except for eigenvalue decomposition. Our experiments were carried out in a cluster of A100 GPUs and use BF16 precision.

Specifically, during the calibration process, we use Pytorch to track the required auto-correlation matrices; this simply done by averaging repeated instantiations of $\mathbf{X}\mathbf{X}^T$ as described in Section 3.

Once the calibration of auto-correlation matrices is over, we use DLPack to transfer them from the Pytorch framework to RAPIDS framework. We then use the CUPY library in RAPIDS to perform fast (a few milliseconds per auto-correlation matrix) eigenvalue decomposition on GPUs. After truncating eigenvectors, we send back the projection matrix to Pytorch using DLPack.

Once the projection matrix $\mathbf{P}$ is calibrated and inference/training is to be done using ESPACE, we simply insert a projection operation within the Megatron implementation to perform the operations in (3) as appropriate. The projection matrices are inserted as Pytorch buffers, rather than parameters, since they do not get updated during training.

## B.2    ESPACE configurations

In Section 4, we mentioned that ESPACE was applied at each layer such that the number of components $L$ satisfies two constraints: (a) be a power of two for best tensor core utilization, and (b) yield a compression of at least 50% at that layer. The exact values of $L$ for each model and layer type are included in Table 2. Note that the only exception corresponds to QKV layers in Llama2-13B and Nemotron4-15B, where we use a value of $L = 2048$ which corresponds to a compression of $\sim 45\%$ instead of $>50\%$ at least. This is only because this amount of compression is already significant that we didn't feel the need to push for $L = 1024$, which would have lead to a compression of $> 70\%$.

## B.3    Retraining hyperparameters

By and large, we use the exact same recipe that was used to pretrain the open source GPT3 models [33]. As mentioned in Section 4, the only modification to hyperparameters is disabling dropout and weight decay, and identical hyperparameters are used for both sets of experiments on GPT3 and Llama2 families. The only arbitrary choices we had to make was on the selection of learning rate schedule and global batch size. We use a cosine decay for all runs, and remaining choicesa are as follows:

- For GPT3-1.3B, the initial learning rate is set to $1.0 \times 10^{-4}$, the final learning rate is set to $1.0 \times 10^{-5}$, and the global batch size is set to 512.
- For GPT3-8B, the initial learning rate is set to $5.0 \times 10^{-5}$, the final learning rate is set to $5.0 \times 10^{-6}$, and the global batch size is set to 512.
- For GPT3-22B, the initial learning rate is set to $5.0 \times 10^{-5}$, the final learning rate is set to $5.0 \times 10^{-6}$, and the global batch size is set to 1024.
- For Llama2-7B and Llama2-13B, training is done in two stages (each of 100B tokens). In the first stage, the initial learning rate is set to $5.0 \times 10^{-4}$, and the final learning rate is set to $5.0 \times 10^{-5}$. In the second stage, the initial learning rate is set to $5.0 \times 10^{-5}$, and the final learning rate is set to $5.0 \times 10^{-6}$. For both stages, the global batch size is set to 256.
- For Nemotron4-15B, the initial learning rate is set to $1.0 \times 10^{-5}$, the final learning rate is set to 0, and the global batch size is set to 512.

We did not perform hyperparameter tuning, the above was purely arbitrary, but based on the following intuition:

Table 2: Number of components $L$ retained by ESPACE for each layer. For models implementing Tensor Parallelism (TP), we apply ESPACE *per rank*.

| Attn QKV layers | Attn Proj layers | FC1 (H-to-4H) layers | FC2 (4H-to-H) Layers |
|---|---|---|---|
| GPT3-1.3B (TP=1) | | | |
| **GEMM Dimension**: | **GEMM Dimension**: | **GEMM Dimension**: | **GEMM Dimension**: |
| $K = 2048$  $N = 6144$ | $K = 2048$  $N = 2048$ | $K = 2048$  $N = 8192$ | $K = 8192$  $N = 2048$ |
| **ESPACE**: $L = 512$ | **ESPACE**: $L = 512$ | **ESPACE**: $L = 512$ | **ESPACE**: $L = 512$ |
| **Compression**: 66% | **Compression**: 50% | **Compression**: 69% | **Compression**: 69% |
| GPT3-8B (TP=4) | | | |
| **GEMM Dimension**: | **GEMM Dimension**: | **GEMM Dimension**: | **GEMM Dimension**: |
| $K = 4096$  $N = 12288$ | $K = 4096$  $N = 4096$ | $K = 4096$  $N = 16384$ | $K = 16384$  $N = 4096$ |
| **GEMM per rank**: | **GEMM per rank**: | **GEMM per rank**: | **GEMM per rank**: |
| $K = 4096$  $N = 3072$ | $K = 1024$  $N = 4096$ | $K = 4096$  $N = 4096$ | $K = 4096$  $N = 4096$ |
| **ESPACE**: $L = 1024$ | **ESPACE**: $L = 256$ | **ESPACE**: $L = 1024$ | **ESPACE**: $L = 1024$ |
| **Compression**: 66% | **Compression**: 69% | **Compression**: 69% | **Compression**: 50% |
| GPT3-22B (TP=8) | | | |
| **GEMM Dimension**: | **GEMM Dimension**: | **GEMM Dimension**: | **GEMM Dimension**: |
| $K = 6144$  $N = 18432$ | $K = 6144$  $N = 6144$ | $K = 6144$  $N = 24576$ | $K = 24576$  $N = 6144$ |
| **GEMM per rank**: | **GEMM per rank**: | **GEMM per rank**: | **GEMM per rank**: |
| $K = 6144$  $N = 2304$ | $K = 768$  $N = 6144$ | $K = 6144$  $N = 3072$ | $K = 3072$  $N = 6144$ |
| **ESPACE**: $L = 2048$ | **ESPACE**: $L = 256$ | **ESPACE**: $L = 2048$ | **ESPACE**: $L = 1024$ |
| **Compression**: 56% | **Compression**: 63% | **Compression**: 58% | **Compression**: 50% |
| Llama2-7B (TP=4) | | | |
| **GEMM Dimension**: | **GEMM Dimension**: | **GEMM Dimension**: | **GEMM Dimension**: |
| $K = 4096$  $N = 12288$ | $K = 4096$  $N = 4096$ | $K = 4096$  $N = 22016$ | $K = 11008$  $N = 4096$ |
| **GEMM per rank**: | **GEMM per rank**: | **GEMM per rank**: | **GEMM per rank**: |
| $K = 4096$  $N = 3072$ | $K = 1024$  $N = 4096$ | $K = 4096$  $N = 5504$ | $K = 2752$  $N = 4096$ |
| **ESPACE**: $L = 1024$ | **ESPACE**: $L = 256$ | **ESPACE**: $L = 1024$ | **ESPACE**: $L = 512$ |
| **Compression**: 66% | **Compression**: 69% | **Compression**: 69% | **Compression**: 69% |
| Llama2-13B (TP=8) | | | |
| **GEMM Dimension**: | **GEMM Dimension**: | **GEMM Dimension**: | **GEMM Dimension**: |
| $K = 5120$  $N = 15360$ | $K = 5120$  $N = 5120$ | $K = 5120$  $N = 27648$ | $K = 13824$  $N = 5120$ |
| **GEMM per rank**: | **GEMM per rank**: | **GEMM per rank**: | **GEMM per rank**: |
| $K = 5120$  $N = 1920$ | $K = 640$  $N = 5120$ | $K = 5120$  $N = 3456$ | $K = 1728$  $N = 5120$ |
| **ESPACE**: $L = 2048$ | **ESPACE**: $L = 256$ | **ESPACE**: $L = 2048$ | **ESPACE**: $L = 1024$ |
| **Compression**: 47% | **Compression**: 55% | **Compression**: 53% | **Compression**: 60% |
| Nemotron4 (TP=8) | | | |
| **GEMM Dimension**: | **GEMM Dimension**: | **GEMM Dimension**: | **GEMM Dimension**: |
| $K = 6144$  $N = 8192$ | $K = 6144$  $N = 6144$ | $K = 6144$  $N = 24576$ | $K = 24576$  $N = 6144$ |
| **GEMM per rank**: | **GEMM per rank**: | **GEMM per rank**: | **GEMM per rank**: |
| $K = 6144$  $N = 1024$ | $K = 768$  $N = 6144$ | $K = 6144$  $N = 3072$ | $K = 3072$  $N = 6144$ |
| **ESPACE**: $L = 2048$ | **ESPACE**: $L = 256$ | **ESPACE**: $L = 2048$ | **ESPACE**: $L = 1024$ |
| **Compression**: 42% | **Compression**: 67% | **Compression**: 58% | **Compression**: 50% |

- For GPT3 models, we use a smaller learning rate for larger models, and start with a learning rate $10\times$ smaller than it's pre-training value. We use identical batch sizes as pre-training.
- For Llama2 models, as pre-training hyperparameters are undisclosed, we use our best guess of what *could* work well. The two stage training approach is inspired by a recent work on 1.58-bit LLMs [49], while the choice of a batch size of 256 is inspired by ChipNemo [50].

## B.4   GEMM latency measurements

Here we describe the methodology employed to measure GEMM latency as reported in Table 1. We assume a batch size of 1, such that the $M$ dimension of tensor $\mathbf{X}$ equals the sequence length (2048 and 4096 for GPT3 and Llama2 models, respectively). We also assume a single-GPU implementation throughout, such that any tensor parallelism is first folded into single GEMM per-layer. Similar to our accuracy experiments, we use BF16 precision for all latency measurements.

For each GEMM layer implementing either (1) or (3), we measure its latency individually. In Table 1, we report aggregated measurements depending on the model configuration. Specifically, we measure

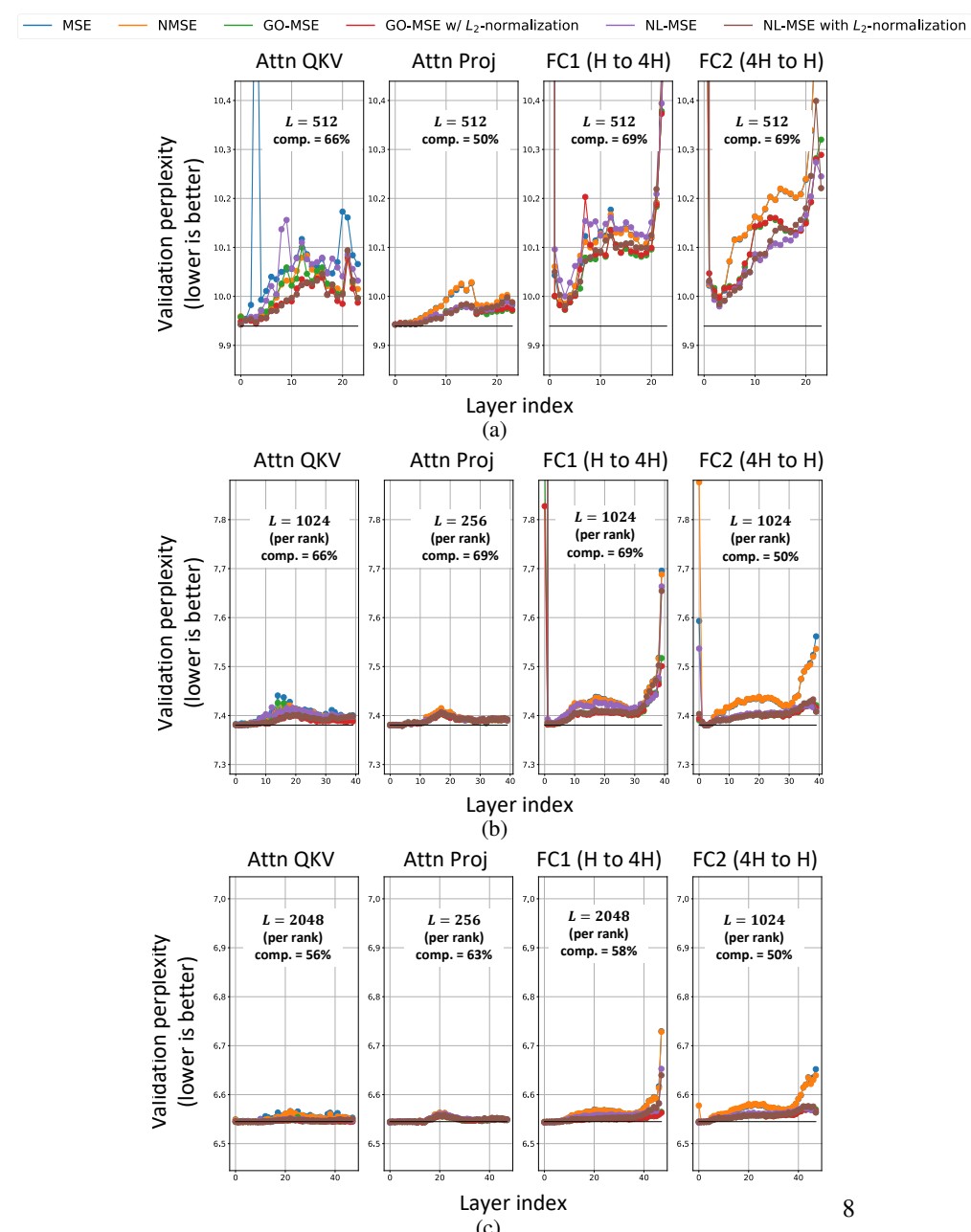

Figure 5: Sensitivity studies on the choice of projection construction for (a) GPT3-1.3B, (b) GPT3-8B, (c) GPT3-22B. For each layer, we apply ESPACE out-of-the-box using the six various candidates for the projection matrix $\mathbf{P}$ constructed in Section 3. The black line corresponds to the baseline perplexity.

latency of computing QKV, Proj, FC1, and FC2 GEMMs with dimensions listed in Table 2, and then add all results together for each transformer block of the corresponding model.

We measure individual GEMM latencies in Pytorch. Specifically, for each configuration, we sample 1000 set of matrices of appropriate dimension and compute the appropriate GEMM. We synchronize before and after the computation occurs, and record times after synchronization. The elapsed times are averaged and then aggregated. Since the measurements were taken with native PyTorch code, we note that the implementation is un-optimized. Further improvements could be possible in future work from removing PyTorch overheads, implementing custom fused kernels, or other optimizations.

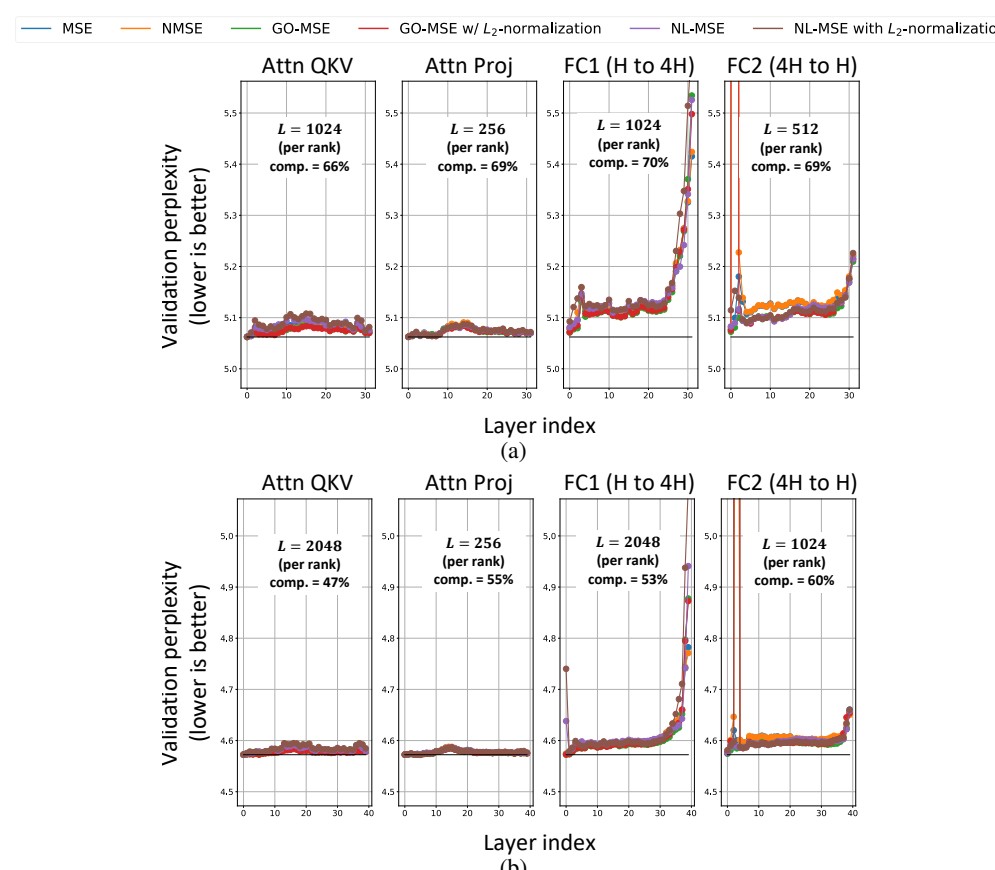

Figure 6: Sensitivity studies on the choice of projection construction for (a) Llama2-7B, (b) Llama2-13B. For each layer, we apply ESPACE out-of-the-box using the six various candidates for the projection matrix **P** constructed in Section 3. The black line corresponds to the baseline perplexity.

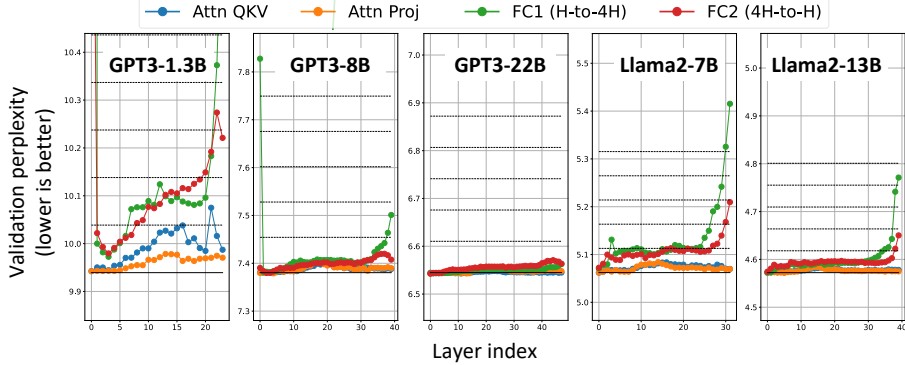

Figure 7: Validation perplexity when ESPACE is applied out-of-the-box one layer at a time using the best calibrated projection matrix **P** as identified by the sensitivity study in Figures 5 and Figures 6. The black line corresponds to the baseline perplexity and the dashed lines correspond to 1% increments over it.

## C   Additional experimental results

In this appendix, we include additional experimental results that were not included in the main paper. These results are not essential to the description of our work nor its conclusion, and the main paper integrally contains all essential information related to our contribution. The additional results listed in this appendix are for the benefit of readers interested in going further and learning about fine-grained details behind the main results of Section 4.

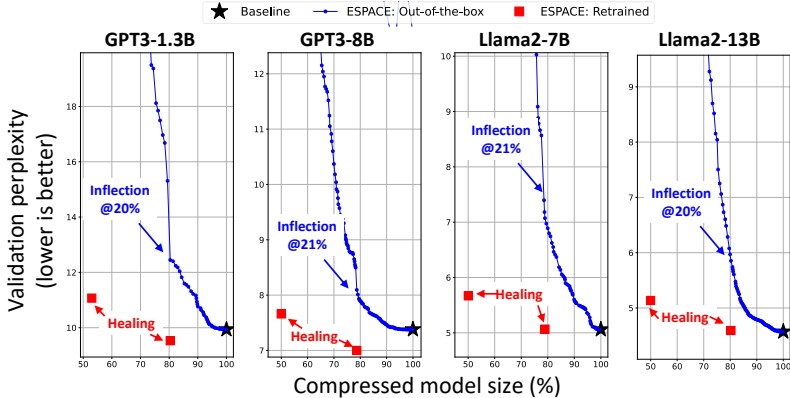

Figure 8: Progressive out-of-the-box application of ESPACE on GPT3{1.3B, 8B} and Llama2-{7B, 13B}. The plot for GPT3-22B was provided in the main text in Figure 3. The progressive application of ESPACE is based on the ranking of layers from least to most destructive based on validation perplexity sensistivity in Figure 7.

## C.1 Sensitivity studies on the construction of projection matrix

In Section 3, we presented theoretical results leading to six choices for the construction of projection matrix $\mathbf{P}$. These constructions were done in a manner to optimize one of the following fidelity metrics: MSE, NMSE, GO-MSE, GO-MSE with $L_2$-normalization, NL-MSE, and NL-MSE with $L_2$-normalization. Here we show the impact of various choices of $\mathbf{P}$ constructions on the validation perplexity, at a per-layer granularity. Specifically, we apply the ESPACE projection out-of-the-box one layer at a time, for each of the six candidates, and evaluate the resulting validation perplexity. In this way, we are able to determine which of the six candidate choices of $\mathbf{P}$ works best at each layer.

The results of this sensitivity analysis are included in Figures 5 and 6 for GPT3 and Llama2 models, respectively. It is shown that the best choice of projection matrix $\mathbf{P}$ depends on layer instance, and there is no clear pattern to find out which solution works best *a priori*. This result justifies the need to optimize several proxy metric for accuracy, not just the MSE of activation approximation. Particularly, most solutions do appear to be related to GO-MSE and NL-MSE, as well as thei $L_2$-normalized variants. Therefore, these results provide supporting evidence on the importance of the results in Proposition 3 and Theorem A.4. This also validates the choice of using bounds on GO-MSE and NL-MSE for optimization since closed form solution for the unbounded metrics are elusive.

## C.2 Progressive application of ESPACE to the layers of a network

Once the best projection matrix $\mathbf{P}$ is identified for each layer, we plot the corresponding validation perplexity for out-of-the-box application of ESPACE at the corresponding layer using the corresponding choice of $\mathbf{P}$. These results are shown in Figure 7, where several observations are made. First, larger models have more resilience to out-of-the-box application of ESPACE; this observation was made in Section 4. Second, it appears that FC1 layers are the most sensitive ones, followed by FC2, and QKV/Proj layers are generally robust to the application of ESPACE. Finally, we observe that in some instances, some layers close to the input and output (i.e., on either ends of the model) appear to be most sensitive to the application of ESPACE. This behavior was observed in other works on compression, such as shortGPT [51].

As mentioned in Section 4, layers are then sorted according to their impact on perplexity from least to most destructive. ESPACE is then progressively applied to out-of-the-box to all layers according to this ranking. In Figure 3 in the main text, we had shown the results corresponding to this applciation for GPT3-22B. In Figure 8, we show similar results for the other four networks we experimented on, i.e., GPT3-{1.3B, 8B} and Llama2-{7B, 13B}. Similar to the findings on GPT3-22B, we do observe an inflection point after which accuracy degradation accelerates. This inflection occurs around 20% for GPT3-{1.3B, 8B} and Llama2-{7B, 13B}. With retraining, the healing process recovers accuracy for all models, as detailed in Section 4.

We note the following:

- For GPT3-1.3B, we exclude some layers from the application of ESPACE. These are the layers for which out-of-the-box application of ESPACE leads to a validation perplexity increase of more than 2% compared to the baseline. These layers can be found in Figure 7 and correspond to several FC1 and FC2 layers close to either ends of the model. For this reason, GPT3-1.3B is compressed to 47% instead of 50% in Section 4 and Table 1.
- For GPT3-22B, we apply ESPACE to all layers since validation perplexity increase is very small in Figure 7. For this reason, the overall compression for GPT3-22B is slightly over 50%; it is 55% in Section 4 and Table 1.

