# OpenReview forum: "ESPACE: Dimensionality Reduction of Activations for Model Compression"
_NeurIPS.cc/2024/Conference — NeurIPS 2024 poster_

### Official Review · Reviewer_1FAL · 2024-06-28

**Soundness:** 3
**Presentation:** 3
**Contribution:** 2
**Rating:** 6
**Confidence:** 4

**Summary:**

The proposed method applies a PCA-inspired method to the activations of LLMs for model compression.

**Strengths:**

- While most papers focus on quantization, pruning, or weight decomposition, the proposed approach goes in an interesting direction.
- The paper is easy to follow and the proposed method is simple.
- The studied problem is relevant.
- The limitations of the method are described.
- The experiments contain multiple ablations to verify the effectiveness of the method.

**Weaknesses:**

- Compression is never well-defined. Is it compression ratio w.r.t. W? What if you do not compress all weights of the model (line 112)?
- Comparison with related baselines is missing. I understand your work is orthogonal to pruning and quantization methods, but it would make sense to compare with methods that use weight decomposition such as ASVD or the ones you mention in the paragraph starting on Line 41.
- The practical benefit of the method can be better emphasized.
- The quality of the figures can be improved, e.g., the text is grainy and Fig. 2 is too small to read.
- Line 192, i==j -> i=j

**Questions:**

- The paper could be made much stronger if you could show some practical benefits in terms of, e.g., inference time, VRAM requirement, or storage requirements. Right now you show compression ratios, but I believe more concrete numbers can make the method much more appealing.

**Limitations:**

The limitations are well addressed.

---

> ### Author Rebuttal · Authors · 2024-08-02
>
> We thank the reviewer for the feedback provided, and for recommending acceptance of our paper. We also appreciate the reviewer underscoring the novelty, clarity, relevance, technicality, and substantiveness of our work. Here we provide answers to the concerns and questions raised:
> * Response to Weakness #1 on the definition of compression ratio.
> * * We apologize for the confusion. We had indeed properly defined compression ratio in the paragraph starting on line 276, but did not provide an explicit formula. The sizes of matrices $\mathbf{W}$, $\mathbf{P}$, and $\mathbf{P}^T\mathbf{W}$ were explicitly stated to be $K\times N$, $L\times K$, and $L\times N$, respectively, at lines 161-166. We should have repeated those when introducing compression ratio at line 276 for better clarity, and this will be rectified in the revision. We also confirm that this compression refers to model size reduction and therefore does not consider layers involving cross-multiplications of activations (line 112). On lines 284-286, we had mentioned that it was beyond the scope of our work to evaluate the impact of ESPACE when taking into account non-GEMM layers. However, we’ve supplemented our results to provide preliminary results on those. Indeed, in the response below, we’ve added time-to-first-token measurements.
> * Response to Weakness #2 on a lack of comparison to SOTA tensor decomposition methods.
> * * We agree with the reviewer that a comparison with other SOTA methods is required to establish a benchmark of ESPACE’s effectiveness. In our submission, we did include this comparison in Figure 4 and section 4.4 (Page 9). Indeed, we compared our Llama2-7b results to those of ASVD, SVD-Lora, and SliceGPT; all contemporary tensor decomposition compression works who have reported perplexity vs compression results for that model. This comparison showed that ESPACE noticeably advances the SOTA. Indeed, while lossless compression cannot be achieved beyond 5% compression (achieved by ASVD) for other methods, ESPACE maintains accuracy with 20% compression. Similarly, the 0.5 perplexity increase at 50% compression with ESPACE matches SliceGPT compression of 10%. Therefore, ESPACE generally advances tensor decomposition compressibility by a factor of 4-to-5x when matching other methods’ accuracy. It seems that this part of our discussion was missed in your original review. We apologize for not highlighting these comparisons more in our manuscript, this will be rectified in the revision.
> * Response to Weaknesses #4 and #5. Thanks! The figures will be made bigger, and the equal sign will be fixed.
> * Response to Weakness #3 and Question #1 on metricizing practicality benefits.
> * * The reviewer is absolutely right about the importance of highlighting practical benefits. We first mention that in the submitted paper, we had included measurements on the speed-up in the matrix multiplication layers. These are included in Tables 1 and 2, and we refer the reviewer to check again if those were missed in the initial review. The findings were that with ~50% compression, ESPACE reduces the latency in GEMM layers by 35%-to-45%. In this rebuttal, per the reviewer’s suggestion, we’ve supplemented these results and measured the time-to-first-token (TTFT) for all models. The experimental setup is similar to that of GEMM latency measurements in Section 4, where we use NVIDIA A100 GPUs, a batch size of 1, and no tensor parallelism. These new results are included below (these results are also included as Table C in the attachment to the common rebuttal):
>
> |Model (Compression)    |Total GEMM Latency (From submission) | TTFT (New result) |
> |-|-|-|
> |Baseline GPT3-1.3B     |24.2ms            |39.8ms       |
> |ESPACE GPT3-1.3B (20%) |20.6ms (-15%)     |36.1ms (-9%) |
> |ESPACE GPT3-1.3B (47%) |15.9ms (-34%)     |31.7ms (-20%)|
> |-|-|-|
> |Baseline GPT3-8B       |136ms             |186ms        |
> |ESPACE GPT3-8B (21%)   |110ms (-19%)      |155ms (-16%) |
> |ESPACE GPT3-8B (50%)   |76.8ms (-44%)     |122ms (-35%) |
> |-|-|-|
> |Baseline GPT3-22B      |354ms             |457ms        |
> |ESPACE GPT3-22B (40%)  |229ms (-35%)      |313ms (-31%) |
> |ESPACE GPT3-22B (55%)  |181ms (-49%)      |261ms (-32%) |
> |-|-|-|
> |Baseline Llama2-7B     |210ms             |368ms        |
> |ESPACE Llama2-7B (21%) |169ms (-19%)      |322ms (-12%) |
> |ESPACE Llama2-7B (50%) |113ms (-46%)      |266ms (-28%) |
> |-|-|-|
> |Baseline Llama2-13B    |406ms             |643ms        |
> |ESPACE Llama2-13B (20%)|336ms (-17%)      |561ms (-13%) |
> |ESPACE Llama2-13B (50%)|259ms (-36%)      |447ms (-31%) |
> |-|-|-|
> * * As we can see, the speed-up in GEMM latency does lead to TTFT reduction. However, since some time is spent in non-GEMM layers (e.g., cross-multiplication of activations in attention), the relative speed-up compared to the baseline is slightly lower for ESPACE. The improvement in TTFT is still significant, and 50% compression with ESPACE leads to 20%-to-35% TTFT reduction. This shows that ESPACE practically improves inference time. In addition, we would like to note that storage requirements are also encapsulated by the previously reported model size reduction achieved via compression. In terms of VRAM requirements, since smaller matrices are loaded to memory, we expect some VRAM requirement reduction. However, a thorough architectural study is needed to quantify accurately what benefits ESPACE can lead. We defer this for future work, but we will include a summary of all of the above, as well as the new TTFT results in the revision. We thank the reviewer, as the above significantly strengthens our work.
>
> Once again, we thank the reviewer for the encouraging comments and productive feedback! We hope our responses above were satisfactory to the reviewer. It would be much appreciated if the reviewer would consider increasing their score.

---

> > ### Comment · Reviewer_1FAL · 2024-08-08
> >
> > Thanks for your rebuttal, it answers my questions. It is indeed nice that the proposed method can reduce GEMM time. I will keep my score.

---

> > > ### Author Response · Authors · 2024-08-08
> > > **Thank you**
> > >
> > > We thank the reviewer for reading our rebuttal, acknowledging our answers, and sharing further positive comments on our results. We kindly ask the reviewer to consider increasing the score in order to reflect their post-rebuttal opinion of our work.

---

### Official Review · Reviewer_ndkj · 2024-07-11

**Soundness:** 3
**Presentation:** 3
**Contribution:** 3
**Rating:** 5
**Confidence:** 4

**Summary:**

The paper introduces ESPACE (Eigen Static Principal Activation Component Estimation), a technique for compressing large language models (LLMs) by focusing on the dimensionality reduction of activation tensors rather than the traditional weight-centric tensor decomposition. The method involves projecting activation tensors onto a pre-calibrated set of principal components, which reduces the dimensionality of activations and results in weight compression during inference through matrix multiplication associativity.

**Strengths:**

The paper introduces ESPACE for compressing LLMs through activation-centric dimensionality reduction. This differs from traditional weight-centric tensor decomposition techniques, which adds novelty to the research.

The authors provide a solid theoretical foundation for their approach, including the derivation of optimal constructions for projection matrices to minimize mean squared error and forward propagated noise metrics.

ESPACE can achieve up to 50% compression of models with a small increase in perplexity and a practical reduction in inference (GEMM) latency.

**Weaknesses:**

To me ESPACE is simply PCA projection (with a reinvented name) of activation tensor. Can the authors defend a bit?

The paper suggests that ESPACE can be used in conjunction with other compression techniques like pruning and quantization, as well as non-compressive acceleration methods such as speculative decoding. However, it does not provide empirical results for these combinations, which makes the claim unconvincing.

Although the authors claim that ESPACE is orthogonal to other compression techniques, it would strengthen the paper to include comparisons with more baselines, such as pruning, to better demonstrate the significance of the proposed method. E.g., when compressed respectively & individually with ESPACE, pruning and quantization to the same model size, how do their perplexities compare to each other? You can demonstrate that on small models like llama-58M or GPT2-97MB.

The necessity for a calibration phase that involves forward-passing multiple batches of data to estimate activation auto-correlation matrices might not scale well with larger datasets or more complex models, potentially limiting the method’s applicability in diverse settings.

**Questions:**

How does the calibration set impact the final performance?

Can you provide a sensitivity profile vs layers in the Transformer model? I.e., how would the local ranks of projection in individual layers affect the global performance? (with the belief that a deep compression in early layers can hamper the output even higher ranks are used for later layers.) A portfolio of layers’ importance would be crucial for the compression strategy.

How does the projection influence the well-known outlier issue in activations and sometimes in the key values, which is observed in lots of LLM works?

**Limitations:**

The calibration set plays a crucial role in the quality of the principal components for projection, this needs to be analysed deeply to warrant good results.

---

> ### Author Rebuttal · Authors · 2024-08-02
>
> We thank the reviewer for the feedback provided, and for recommending acceptance of our paper. We also appreciate the positive comments highlighting the novelty of our approach, its solid theoretical foundation, and the promising empirical results. Here we provide answers to the concerns and questions raised:
> * Response to Weakness #1 on the similarity to PCA.
> * * We are happy to defend: we were indeed inspired by this classic algorithm and in fact did provide a comparison to PCA (lines 200-204). PCA’s goal is to extract low dimensional features having maximum correlation with input data. In contrast, ESPACE is derived such that the compression in eq. (3) is achieved while minimizing various accuracy metrics, which were theoretically derived in Section 3. Crucially, ESPACE leverages ergodicity of activation autocorrelation (see Section 3.1) to produce static projections enabling compression as described in eq (3) and Figure 2. This would not be feasible using PCA, as extracting low dimensional features of activations on the fly would require an online algorithm, which is expensive. In the revised manuscript we will further defend these differences between ESPACE and PCA. We thank the reviewer for bringing this up.
> * Response to Weaknesses #2 and #3 on comparisons between ESPACE and other compression techniques such as quantization or pruning.
> * * We have mentioned that ESPACE can be applied with other techniques because its implementation is orthogonal to others. For instance, ESPACE can be implemented using a quantized or sparse number format. However, while that is the case, we did mention compression fundamentally makes models more susceptible to noise and that a detailed study of combining ESPACE with other techniques was left for future work (Section 1.1; line 33-35). This claim of orthogonality will be toned down in our updated manuscript. With that said, we appreciate the suggestion to include studies with other compression techniques. As the reviewer appreciates, it requires time and efforts to generate such results. But for this rebuttal, we have studied the following: ESPACE and FP8 quantization. While sub-8-bit quantization has been explored, we chose FP8 quantization because it is popular among practitioners thanks to its versatility: it usually does not require QAT and is available in NVIDIA Hopper GPUs. Let us compare the following: baseline BF16, ESPACE BF16, baseline FP8, and ESPACE FP8. We use Hopper FP8 (E4M3) and employ per-tensor dynamic max scaling. Both weights and activations are quantized to FP8, and for ESPACE, additional tensors in eq (3) are quantized to FP8, i.e., projected activation, projection matrix, and precomputed weight-projection product. For brevity, we employ GPT3-8B and Llama2-7B as representative models. Complete results with all other models studied will be included in the revision. We evaluate Wikitext test perplexity; our results are as follows (these results are also included as Table B in the attachment to the common rebuttal):
>
> |Model (Compression)    |BF16 |FP8 |
> |-|-|-|
> |Baseline GPT3-8B       |7.38 |7.41|
> |ESPACE GPT3-8B (21%)   |7.00 |7.22|
> |ESPACE GPT3-8B (50%)   |7.66 |7.85|
> |-|-|-|
> |Baseline Llama2-7B     |5.06 |5.08|
> |ESPACE Llama2-7B (21%) |5.07 |5.13|
> |ESPACE Llama2-7B (50%) |5.67 |5.80|
> |-|-|-|
> * * It turns out that with ESPACE, susceptibility to quantization noise is slightly worse, where perplexity increases by ~0.1-0.2. This is reasonable since a compressed model is expected to be less robust to quantization noise (our prediction from lines 33-35). We thank the reviewer for the suggestion to add such results making our work stronger. We concede that ESPACE does makes quantization a bit harder, although we emphasize that the above are simple PTQ tests which could be improved with further optimizations such as SmoothQuant, GPTQ, or even QAT.
> * Response to Weakness #4, Question #2, and Limitation #1 on the impact of calibration set choice.
> * * We have found that the performance was not sensitive to the choice of calibration set. Specifically, we use 512 random sequences for calibration but found that consistent results were obtained when using smaller sets, e.g., 32 sequences (experiments on varying calibration set size were done for the GPT3-1.3B model). Thus, we believe calibration to be robust to the choice of calibration dataset.
> * Response to Question #2 on layer-wise sensitivity profiles.
> * * Layer-wise sensitivities to projections are included in detail in Appendix C, where the exact question is being answered for each layer of each model studied. The portfolio of layers’ importance was indeed utilized when selecting the compression configuration setting (see Section 4). To meet the page limit, we elected to keep the sensitivity study in Appendix C but did refer to it in the main text in Section 4.2.  We ask the reviewer to check that these parts of the submission did in fact provide answers to the reviewer’s question.
> * Response to Question #3 on the outlier issue.
> * * Since ESPACE is an algebraic compression technique, as opposed to numerical ones such as quantization and pruning, we have not had any issue with the presence of outliers. Also recall that our experiments use BF16 precision which could be shielding ESPACE from the outlier issue. Nevertheless, we are encouraged that ESPACE has circumvented the outlier issue, but we will need further studies to make definite claims, particularly when using low-precision quantization. So, this is a good direction for future work which blends naturally with our plans. Thank you!
>
> Once again, we thank the reviewer for the encouraging comments and productive feedback! We hope our responses above were satisfactory to the reviewer. It would be much appreciated if the reviewer would consider increasing their score.

---

> > ### Comment · Reviewer_ndkj · 2024-08-10
> > **Thanks for your reply!**
> >
> > I am happy with the authors' reply and the additional experimental results. I tend to keep my score.

---

> > > ### Author Response · Authors · 2024-08-10
> > > **Thank you**
> > >
> > > We are grateful for the reviewer's acknowledgment of our rebuttal and positive opinion towards our replies and additional experimental results. We kindly ask the reviewer to consider increasing the score in order to reflect their post-rebuttal opinion of our work.

---

### Official Review · Reviewer_maTu · 2024-07-12

**Soundness:** 3
**Presentation:** 3
**Contribution:** 3
**Rating:** 6
**Confidence:** 4

**Summary:**

The paper introduces a novel technique for compressing large language models (LLMs) by reducing the dimensionality of activation tensors. The ESPACE method differs from traditional weight-centric compression approaches by focusing on activation tensors instead. ESPACE projects these activations onto a pre-calibrated set of principal components, preserving expressivity during retraining while enabling weight compression at inference through associative matrix multiplication.

**Strengths:**

1. The paper introduces a novel approach to compressing large language models by focusing on the dimensionality reduction of activation tensors rather than the weights themselves. This method, termed ESPACE (Eigen Static Principal Activation Component Estimation), represents a creative combination of principles from tensor decomposition and matrix multiplication associativity.

2. This paper is highly technical and presents a robust foundation for constructing optimal projection matrices that minimize the mean squared error and forward-propagated noise metrics.

3. The paper is well-written and organized logically. The authors effectively communicate complex ideas, making the novel approach accessible to readers.

**Weaknesses:**

1. More direct comparisons with other state-of-the-art activation-based and tensor decomposition methods could provide a clearer benchmark of ESPACE's effectiveness.

2. The paper could benefit from a more thorough discussion of the trade-offs in selecting compression settings and projection matrices for different application scenarios.

3. Including a robust error analysis would help identify potential weaknesses or failure modes of ESPACE, enhancing understanding of its limitations.

4. The study focuses primarily on GPT3 and Llama models, potentially limiting the generalizability of the results to other architectures.

5. The paper does not extensively explore ESPACE's performance across various inference tasks that differ in length or complexity, which could provide a deeper understanding of its practical implications.

**Questions:**

1. How does ESPACE perform on different types of inference workloads, particularly those requiring higher precision or longer inference times? Additional details on these scenarios would help assess the method’s practical applicability in diverse real-world settings.

2. Could you elaborate on the decision-making process for selecting specific compression settings and projection matrices? A detailed discussion on how to balance compression rate and model performance in various contexts would be highly valuable for practitioners.

3. Have you conducted any tests to evaluate ESPACE’s robustness under unusual conditions or edge cases? Sharing any findings or future plans for such analyses would provide a clearer understanding of the method’s limitations and potential areas for improvement.

**Limitations:**

1. Include evaluations or discuss plans for testing ESPACE on various model architectures to better understand its generalizability.

2. Discuss potential challenges and solutions for deploying ESPACE in real-world systems, focusing on real-time performance and integration with existing workflows.

---

> ### Author Rebuttal · Authors · 2024-08-02
>
> We thank the reviewer for the feedback provided, and for recommending acceptance of our paper. We also appreciate the positive comments provided with respect to the novelty, technicality, and clarity of our work. Here we provide answers to the concerns and questions raised:
> * Response to Weakness #1 on a lack of comparison to SOTA tensor decomposition methods .
> * * We agree with the reviewer that a comparison with other SOTA methods is required to establish a benchmark of ESPACE’s effectiveness. In our submission, we did include this comparison in Figure 4 and section 4.4 (Page 9). Indeed, we compared our Llama2-7b results to those of ASVD, SVD-Lora, and SliceGPT; all contemporary tensor decomposition compression works who have reported perplexity vs compression results for that model. This comparison showed that ESPACE noticeably advances the SOTA. Indeed, while lossless compression cannot be achieved beyond 5% compression (achieved by ASVD) for other methods, ESPACE maintains accuracy with 20% compression. Similarly, the 0.5 perplexity increase at 50% compression with ESPACE matches SliceGPT compression of 10%. Therefore, ESPACE generally advances tensor decomposition compressibility by a factor of 4-to-5x when matching other methods’ accuracy. It seems that this part of our discussion was missed in your original review. We apologize for not highlighting these comparisons more in our manuscript, this will be rectified in the revision.
> * Response to Weaknesses #2 and #3 on application-specificness and robustness analyses.
> * * We appreciate the feedback on discussing application-specific compression and robustness analyses. These were beyond the scope of our work but we will add discussions for these topics in our revised manuscript, and specify directions for future work in these scopes.
> * Response to Weakness #4 on the use of GPT3 and Llama2 and the impact on generalizability.
> * * The reviewer is correct in highlighting the importance of generalizability of results. GPT3 and Llama are contemporary open-source and popular LLM architectures used broadly in various applications. This is why we elected to cover them in our paper. Furthermore, by virtue of having several model instances in these families, we did experiment on five models overall (GPT3-1.3B, GPT3-8B, GPT3-22B, Llama2-7B, Llama2-13B). Thus, our experiments do cover a wide range of model sizes. Nevertheless, we agree that it is desirable to cover more model architectures, and this is absolutely in our plan for future work, which we plan to discuss in the revision.
> * Response to Weakness #5 on the variety of tasks studied.
> * * We have evaluated ESPACE on a wide range on inference tasks, including several LM evaluation harness benchmarks (see Tables 1 and 2), as well as the complex MMLU benchmark (which we added as part of this rebuttal in response to Reviewer yYTB above). But the reviewer is correct that we have not studied specifically the issue of varying context length. This is a good direction for future work, which will be mentioned in our revision. Thank you!
> * Response to Question #1 on the impact of high precision and long inference times.
> * * This is a great question! With regards to precision, and as mentioned in our paper, we have used BF16 throughout our experiments and never had any numerical issues. Having said that, because our compression technique is algebraic, rather than numeric (e.g., such as works doing quantization and/or pruning), we do not anticipate issues should higher precision be required. With regards to workloads requiring long inference times, such scenarios are specifically instances where ESPACE’s benefits can be leveraged. Indeed, as shown in our paper, ESPACE leads to a reduction in 35-to-45% in GEMM latency. Thus, if GEMM latency is reduced, inference can run faster. In fact, we have recently generated end-to-end inference latency measurements to back this up. Please see our response to reviewer 1FAL on measured time-to-first-token: ESPACE helps significantly speed-up inference.
> * Response to Question #2 on the decision-making process for selecting specific compression settings and projection matrices
> * * We have used validation studies as described in Section 4.2 and further in-depth elaborated on in Appendix B. For the interested practitioner, we also summarize our strategy for these selections here. By using a validation set (which is mutually exclusive to training and test sets), we perform layer-wise studies where we analyzed various compression setting and projection matrix choice at each layer independently. We rank the accuracy profile of various candidate settings for each layer, and subsequently select the best choice to be further experimented on. Specifically, these become the compressed model configurations used for ESPACE in the experimental sections 4.3 and 4.4 in our paper.
> * Response to Question #3 on tests to evaluate ESPACE’s robustness under unusual conditions or edge cases
> * * We have not conducted such studies. We kindly ask the reviewer to provide examples of what qualifies as unusual conditions and edge cases. We will study those as future work and include a relevant discussion in the revision.
>
> Once again, we thank the reviewer for the encouraging comments and productive feedback! We hope our responses above were satisfactory to the reviewer. It would be much appreciated if the reviewer would consider increasing their score.

---

### Official Review · Reviewer_yYTB · 2024-07-13

**Soundness:** 1
**Presentation:** 2
**Contribution:** 3
**Rating:** 5
**Confidence:** 4

**Summary:**

This paper introduces ESPACE, a method that reduces activation in models via tensor decomposition, thereby aiding in the reduction of model size and GEMM latency.

**Strengths:**

- The paper is easy to follow;
- The method is simple but efficient.

**Weaknesses:**

- Wikitext-103 is a specific type of language dataset that focuses on knowledge-based content. However, models like GPT-3 and Llama2 are capable of handling a broader range of tasks, including math and question-answering. Therefore, it is necessary to evaluate perplexity (PPL) on a more diverse set of data.
- ESPACE requires ongoing training. It is important to understand the duration required for ESPACE's continual training process.
- In Table 1 and Table 2, since ESPACE is continually trained with more data than the original model, a direct comparison may not be entirely fair. It would be more convincing to compare the original model after it has been trained on the same amount of data as ESPACE.
- For generative models such as GPT-3 and Llama2 discussed in this paper, there is a need for more comprehensive performance validation, including metrics like mmlu and math benchmarks.

**Questions:**

Please see the weaknesses.

**Limitations:**

The authors address the limitations.

---

> ### Author Rebuttal · Authors · 2024-08-02
>
> We thank the reviewer for the feedback provided. We provide detailed answers to the reviewer below:
>
> * Response to weaknesses #1 and #4 on Wikitext being an insufficient benchmark for empirical results and the need for a more comprehensive set of validation metrics.
>
> * * The reviewer is correct in that Wikitext is a language dataset focusing on knowledge-based content. As such, evaluating Wikitext perplexity is not the only metric to report on for proper evaluation of LLMs. In our submission, we had included evaluations of the models on the diverse LM evaluation harness suite of tasks (BoolQ, Hellaswag, Piqa, Race, Winogrande – see Tables 1 and 2 in Section 4). Of these question-answering tasks, BoolQ and Race are knowledge-based, while Hellaswag, Piqa, and Winogrande comprise common-sense and logical reasoning. Our results (in the submission) for these tasks showed that the accuracy-preserving strength of ESPACE is not limited to Wikitext perplexity, since the scores obtained on these tasks were very close to those of the baseline. In case the reviewer missed the results on downstream task accuracy in the original review, we kindly ask to check again.
>
> * * We’ve further expanded these results by adding MMLU evaluation as per the reviewer’s recommendation. The following table (also included as Table A in the common rebuttal attachment) summarizes these results on the MMLU benchmark (which itself comprises a broad range of tasks including mathematical and logical reasoning in addition to human knowledge). Here we use the same models, compression settings, and checkpoints as those discussed in our submission. MMLU inference is done using the zero-shot approach:
>
>
> |Model (Compression)    |MMLU score|
> |-|-|
> |Baseline GPT3-1.3B     |     25.5%|
> |ESPACE GPT3-1.3B (20%) |     25.3%|
> |ESPACE GPT3-1.3B (47%) |     25.1%|
> |-|-|
> |Baseline GPT3-8B       |     26.3%|
> |ESPACE GPT3-8B (21%)   |     31.5%|
> |ESPACE GPT3-8B (50%)   |     27.4%|
> |-|-|
> |Baseline GPT3-22B      |     36.3%|
> |ESPACE GPT3-22B (40%)  |     42.2%|
> |ESPACE GPT3-22B (55%)  |     39.8%|
> |-|-|
> |Baseline Llama2-7B     |     42.2%|
> |ESPACE Llama2-7B (21%) |     39.6%|
> |ESPACE Llama2-7B (50%) |     32.7%|
> |-|-|
> |Baseline Llama2-13B    |     52.9%|
> |ESPACE Llama2-13B (20%)|     49.4%|
> |ESPACE Llama2-13B (50%)|     39.4%|
> |-|-|
>
> * * As we can see, the results are generally consistent with our earlier findings from our experiments on Wikitext and the set of downstream tasks from the LM evaluation harness that were included in the original submission. We note that MMLU is a complex benchmark, such that the GPT3-1.3B model is too weak to produce meaningful results beyond a random guess. For larger GPT3-models, we once again find that moderate compression using ESPACE leads to improvements over the baseline, specifically for GPT3-8B with 21% compression and GPT3-22B with 40% compression. The improvements in MMLU scores are significant, and even higher than for the benchmarks covered in our submission. Indeed, ESPACE leads to a ~5% absolute increase in MMLU score, which corresponds to a 16%-to-19% relative improvement. On the other hand, 50% compression with ESPACE leads to an accuracy comparable to the baseline, which is consistent with our findings in our original submission. Our results for Llama2 models are not as strong, and we attribute this to the handicaps of our training sessions that were discussed in Section 4.4. Specifically, our experimental setup did not include the pre-training dataset and hyperparameter selection used by Meta to produce accurate Llama2 models. As such, while ESPACE does retain some MMLU accuracy, it does not perform as strongly for Llama2 as it does for GPT3. We also note that the MMLU benchmark is known to be challenging for base (unaligned) models (which we use throughout our studies). Overall, we are glad to include these additional results in our paper as we believe, in agreement with the reviewer, that they improve our work. Thank you for the suggestion!
>
>
> * Response to weakness #2 on the duration of the continual training process.
>
> * * In the original submission, we discussed training duration in Section 4.3 (330B tokens for GPT3) and section 4.4 (200B tokens for Llama2). The reviewer may have missed this discussion in the original review. We ask the reviewer to check again.
>
> * Response to weakness #3 on a direct comparison between compressed model and baseline not being fair due to the consumption of additional data in the continual training process.
>
> * * The reviewer makes an excellent point! Continued training consumes more tokens overall compared to the pre-trained baseline, therefore it is important to understand the impact of the continuous training session and dissociate it from the adaptation to compression. We would like to point out that this is exactly why we chose to retrain the Llama2-7b baseline using the same tokens and training session as those used for the ESPACE models. Indeed, see Table 2 (second row), where we included results for a retrained Llama2-7B model and the motivation for it at line 343 which is specifically to address the concern raised by the reviewer. This retrained baseline now matches the amount of training data as ESPACE, while its accuracy closely matches the original baseline. Therefore, this result allows us to compare ESPACE models to the baselines, either original or retrained ones (since they have matching accuracy).  We kindly ask the reviewer to check this again.
>
> In conclusion, we first acknowledge that the reviewer has made very good points. At the same time, the concerns raised by the reviewer were in fact already addressed in our original submission. In our revision, we will highlight the above points to be more explicit. Nevertheless, we hope the reviewer will consider increasing their score.

---

> > ### Comment · Reviewer_yYTB · 2024-08-12
> >
> > Thanks for the response. The MMLU results of Llamma-2 are still weak as a significant performance reduction. I suggest using SFT data (like Alpaca) to improve the results to validate the possibility of ESPACE to improve the performance.
> >
> > As most of the weaknesses are addressed, I am willing to raise my score. However, I expect the authors to show better empirical results in the next version.

---

> > > ### Author Response · Authors · 2024-08-12
> > > **Thank you!**
> > >
> > > We sincerely thank the reviewer for acknowledging our rebuttal and updating their review to recommend acceptance. We also agree with the reviewer on the need to further improve the Llama2 results and are grateful for the suggestion to use SFT data such as Alpaca. We promise to work on this and show improved results in the revision.

---

### Author Rebuttal · Authors · 2024-08-02

Dear reviewers,
We would like to thank you for the useful feedback to our work. We’ve addressed every concern and question raised in the individual responses. In this common response we would like to emphasize a few points.

First, it seems that some parts of the paper were missed by some reviewers. We wanted to emphasize the presence of the following information in our original submission:

* Evaluations on a diverse set of downstream tasks from the LM evaluation harness (BoolQ, Hellaswag, Piqa, Race, Winogrande – see Tables 1 and 2). In fact, reviewer 1FAL explicitly states our “multiple ablations” as a strength of our work. However, Reviewer yYTB appeared to have missed those and mentioned we only evaluated our method on the Wikitext dataset.

* Information about training time can be found in Section 4 of our submission. Reviewer yYTB inquired about that.

* Results with a retrained Llama2 baseline (see Section 4.4. and Table 2) were included to address the issue of comparing a continuously trained model to a baseline having consumed the same amount of training data. Reviewer yYTB raised this important point but appears to have missed our inclusion of this result in our submission.

* Comparison to SOTA tensor decomposition works   is present in Section 4.4 and Figure 4. This was inquired by Reviewers maTu and 1FAL.

* Studies on layer-wise sensitivities are discussed in-depth in Appendix C and referenced in Section 4.2. This was inquired by Reviewer ndkj.
* A study on inference time proxied by GEMM latency measurements is included in Section 4 and Tables 1 and 2. This was inquired by Reviewer 1FAL.

More detailed responses to the above are provided in the individual rebuttals. We want to emphasize this in case any reviewer had provided lower scores due to missing any of the above information while reading our submission. We hope that the above clarifications can rectify that. In addition, in the individual rebuttals, we provide answers to all other concerns and questions raised by the reviewers.
We also wish to point out that, as part of the rebuttal, and thanks to some excellent points made by the reviewers, we have added three sets of results as follows (these results are included in the attached extra document and as inline tables in the individual responses to make the reading of rebuttals smoother):

* Additional results evaluating all models on the MMLU benchmark. For more details, please refer to the rebuttal to Reviewer yYTB and Table A in the attached document.

* Additional results combining ESPACE with FP8 quantization. For brevity, in the rebuttal we only included such results on GPT3-8B and Llama2-7B as representatives to all models studied. For more details, please refer to the response to Reviewer ndkj and Table B in the attached document.

* Additional latency measurements on the time-to-first-token for all models to supplement the GEMM latency measurements that were present in the original submission. For more details, please refer to the rebuttal to Reviewer 1FAL and Table C in the attached document.

The above set of new results significantly strengthens our work and will be included in the revision. We thank the reviewers again for the feedback.

---

### Decision · Program_Chairs · 2024-09-25

**Decision:**

Accept (poster)

**Comment:**

Although barely above borderline, all reviewers supported acceptance, particularly after the authors' extensive responses clarified or improved some issues. So I'm happy to recommend acceptance.

The idea of compressing the activations by multiplying and reprojecting times a fixed matrix seems very similar to the idea of matrix sketches, and perhaps that is essentially what is going on here. Can the authors comment on this in the final version of the paper?